



# Flexible parallel implicit modelling of coupled Thermal-Hydraulic-Mechanical processes in fractured rocks

Mauro Cacace[1] and Antoine B. Jacquey[1,2]

[1]GFZ - German Research Centre for Geosciences, Potsdam, Germany
[2]RWTH Aachen University, Aachen, Germany

*Correspondence to:* Mauro Cacace (mauro.cacace@gfz-potsdam.de)

**Abstract.** Theory and numerical implementation describing groundwater flow and the transport of heat and solute mass in fully saturated fractured rocks with elasto-plastic mechanical feedbacks are developed. In our formulation, fractures are considered as being of lower dimension than the hosting deformable porous rock and we consider their hydraulic and mechanical apertures as scaling parameters to ensure continuous exchange of fluid mass and energy within the fracture-solid matrix system. The coupled system of equations is implemented into a new simulator code that makes use of a Galerkin Finite Element technique. The code builds on a flexible, object oriented numerical framework (MOOSE, Multiphysics Object Oriented Simulation Environment) which provides an extensive scalable parallel, implicit coupling to solve for the multiphysics problem. The governing equations of groundwater flow, heat and mass transport and rock deformation are solved in a weak sense (either by classical Newton-Raphson or by Free Jacobian Inexact Newton Krylow schemes) on an underlying unstructured mesh. Non-linear feedbacks among the active processes are enforced by considering evolving fluid and rock properties depending on the thermo-hydro-mechanical state for the system and the local structure, i.e. degree of connectivity, of the fracture system. A suite of applications is presented to illustrate the flexibility and capability of the new simulator to address problems of increasing complexity and occurring at different spatial (from centimeters to tens of kilometers) and temporal scales (from minutes to hundreds of years).

## 1 Introduction

Reliable predictions of reservoirs' performances, whether for geothermal and/or fossil energy extraction, or water, $CO_2$ and heat storage rely on an accurate representation of the physical processes responsible for groundwater flow, heat and solute mass transfer, mechanical deformation of the rock solid skeleton, and, ultimately, chemical feedbacks from fluid-rock interactions (Stephansson et al., 2004). All of these processes occur in a naturally complex geological setting, comprising discrete heterogeneities as faults and fractures, span a relatively large spectrum of temporal and spatial scales and interact in a highly non-linear fashion. In natural and engineered systems, onset conditions and the evolution in time and space of a particular process are affected by the initiation and evolution of all the other processes. Therefore, monitored variations of rock masses to natural and anthropogenic perturbations cannot be fully reconciled by considering the causative processes independently. This is particularly the case for reservoir applications which require a complete understanding of the multi-component (fractured)

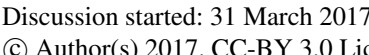



porous rock-fluid system and its multiphysics dynamics to predict the behaviour of a particular reservoir so to improve its productivity and sustainability.

To recognise that the majority of geological observations are the results of non-linear interactions among classes of physical processes has justified the recent development and massive use of so-called multiphysics numerical simulators as comple-

mentary tools to classical experimental and theoretical analyses. Numerical simulations are particularly adapted to the study of a multi-component physical system since they allow a systematic analysis of the dynamics of single processes and their interactions (i) under proper time and length scales, (ii) within complex geometries, and (iii) also considering changing loading conditions. In addition, given their predictive capabilities, numerical modelling techniques can assist studies aiming at estimating performances and potential risks related to different operational scenarios (e.g.Blöcher et al. (2015)).

Numerical modelling of fractured reservoirs, whether petroleum or geothermal, has a long history dating back several decades. Starting from the earlier works in the 1970's, a number of codes have been implemented under various licenses by the modelling community (Jing, 2003; Stephansson et al., 2004). The majority of these codes relies on modelling techniques that were developed in the late 1980's and early 1990's, and that can be grossly subdivided into two classes if based on the approach followed to model coupled processes.

A first class of approaches, referred to as sequential/explicit coupling or operator splitting approaches, relies on splitting the coupled physical problem into classes of possibly linear sub-problems and to numerically solve for each process sequentially. During each time step, the coupling among the processes is enforced by passing input/output data among the respective sub-problems. Usually, external iterations are adopted for simulations characterised by a high degree of non-linearity (Kempka et al., 2016; Chabab and Kempka, 2016). In some cases, sequential coupling is achieved by relying on different simulators,

as for example by coupling a flow simulator to a mechanical simulator for a thermo-hydro-mechanical problem (THOUGH-FLAC family of codes, e.g. Rutqvist (2011)). The main advantage of such approaches stems from numerically integrating relatively complex problems within limited computer resources. However, sequential coupling schemes have important impacts on the efficiency, stability, and accuracy of the numerical solutions. They usually introduce a splitting error in the numerical approximation, which requires a careful monitoring of the non-linear residuals within each single time step (Jha and Juanes,

2007; Preisig and Prévost, 2011). This aspect limits sequential approaches to what are generally referred to as "loosely coupled problems", and they show a relatively slow, if at all, convergence rate for tightly coupled problems. In addition, conditional stability of semi-implicit approaches impose severe time step restrictions thus increasing computation times.

As an alternative, it is possible to solve the full system of coupled equations simultaneously, via a fully implicit coupling approach. This requires to solve for all the variables of the problem simultaneously within an iterative approach, either (in)exact

Newton or simpler Picard methods, for the resulting algebraic system (Knoll and Keyes, 2004). Simultaneous solution schemes demand higher computer storage and processing times to compute (allocate the memory and fill) the Jacobian of the problem than explicit approaches. However, they guarantee higher stability for strongly non-linear problems even with relative large time step sizes (Kim et al., 2015). In addition, recent advances in developing block-type preconditioners and "super convergent" methods for systems of non-linear equations at reduced memory consumption, e.g. preconditioned Jacobian free methods,

make these approaches competitive for field scale applications (Knoll and Keyes, 2004).





Despite all of this, only few attempts have been made so far in developing and implementing fully implicit numerical solutions for THM problems.

The FRACON code by Nguyen and Selvadurai (1998) did consider a monolithic small strain and non-isothermal implementation, but thermal feedbacks effects on the skeleton deformation and pore fluid pressure are only explicitly integrated. The

open source, object oriented project OpenGeosys (Kolditz et al., 2012) does provide a parallel platform for implicit solution of multiphysics problems (Blöcher et al., 2015), but so far, to the authors' knowledge has never been applied to 3D THM problem in fractured reservoirs. More recently, Sun (2015) developed a monolithic framework for solving coupled THM processes at finite strain. However, such an approach has only been applied on generic and simplistic models and lacks a description of complex geological geometries such as fractures and faults.

The goal of this paper is not to summarise the state of the art computational methods for problems that are relevant for reservoir applications. Our interest herein is rather toward computational reliability and performances when simulating the behaviour of a particular reservoir in a way that can be of help to improve scenario-oriented analysis of such systems. In this context, we address issues related on how (i) to quantify the non-linear feedbacks among the different physical processes, and (ii) to represent into a computational model the porous rock-fracture-fluid system by capturing its discontinuous, anisotropic,

inhomogeneous and non-elastic nature (Hudson and Harrison, 1997). At this purpose, we give an overview of the methods implemented into a novel, yet robust and efficient multiphysics and multi-component porous media open source modelling simulator called GOLEM which can deal with all these aspects. Our emphasis throughout this manuscript is to simulate thermal-hydraulic-mechanical (THM) processes of relevance for hydrothermal and geothermal systems. Though GOLEM can also simulate the transport of non-reactive chemical species, we do not discuss this aspect in the present manuscript. At the

same time, to consider additional chemical (fluid-solid) interactions is the subject of future work.

The remaining of the manuscript is structured as follows. In Section 2, we introduce the constitutive model for THM processes for a two phase system consisting of a deformable solid skeleton and fully saturated pores in the presence of discrete heterogeneities as represented by faults and fractures. Attention is given to include the important non-linear feedbacks among the different processes as well as structure-property relationships. These latter include links between fluid properties (e.g. den-

sity and viscosity) and problems variables (e.g. pressure and temperature), as well as evolution of material properties as a function of variations in the state of the reservoirs (e.g. porosity and permeability relationships). In Section 3, the numerical implementation of the derived constitutive model is described. In Section 4, we apply the simulator to a suite of applications of different level of complexity (from synthetic benchmark cases to prototypical reservoir simulations) both in terms of the coupling among the processes considered as well as in terms of the geometry of the natural system. A field application related

to modelling the hydromechanical response of a sandy reservoir during a cyclic injection test will be presented in a separated paper (Jacquey et al. (under review)).





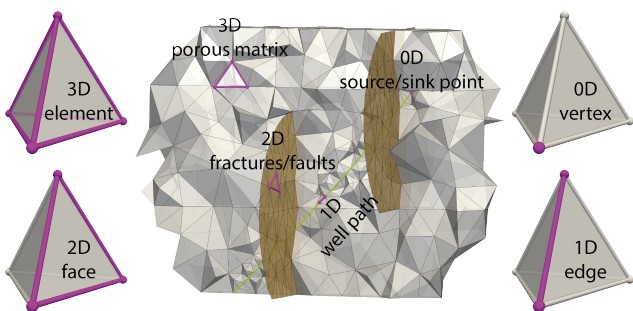

**Figure 1.** Schematic representation of the approach to model fracture rock masses as based on a superposition of lower and higher dimensional geometric elements, after Cacace and Blöcher (2015): 0D vertex (point-like sources/sinks), 1D edge (well paths), 2D face (fractures an faults) and 3D element (porous matrix).

## 2 Mathematical formulation of the problem

In this section, we present the constitutive model for a porous medium consisting of a deformable hosting rock and a mobile liquid saturating its pores. In the following we make use of subscript $f$ and $s$ to refer to the solid and fluid phase respectively. Deformation of the solid skeleton is considered in terms of Biot consolidation theory (Biot, 1956; Biot and Willis, 1957; Biot,

1973) under non-isothermal loading (Geertsma, 1957; McTigue, 1986) and its extension to take into account time and rate dependent inelastic behaviour. The model assumes the existence of a Reference Elementary Volume (REV) for the porous medium in which the two components can interact thermo-hydro-mechanically (Bear, 1988). The resulting physical system therefore incorporates non-isothermal flow of the fluid phase (namely water) within a porous rock which is free to deform (in)-elastically and in the presence of discontinuities as represented by discrete fractures.

Fractures are considered as lower dimensional elements embedded in the porous matrix filled with fluid, see Fig. 1. We consider the length of the discrete fracture as a representative measure of the REV of the porous matrix. This allows to make use of a continuum approach to represent the porous medium. We also take into account the effects of micro-defects (fissures and micro cracks) on the thermal, hydraulic and mechanical behaviour of the porous rock as they mainly affect the evolution of the system material properties (e.g. porosity, see Fig. 2).

In the following the main governing equations are derived. The problem variables considered are the pore fluid pressure $p_f$, the temperature $T$, and the solid displacement vector $\boldsymbol{u}$. Pore pressure is defined as compression positive for water, while stress is defined as tension positive for the solid phase.

The mass balance equation for a deformable, saturated porous medium is described in terms of volumetric averaged mass conservation equations for the fluid and solid phases. Mass conservation therefore requires for the liquid phase,

$$\frac{\partial (n\rho_f)}{\partial t} + \nabla \cdot (n\rho_f \boldsymbol{v}_f) = Q_f \tag{1}$$





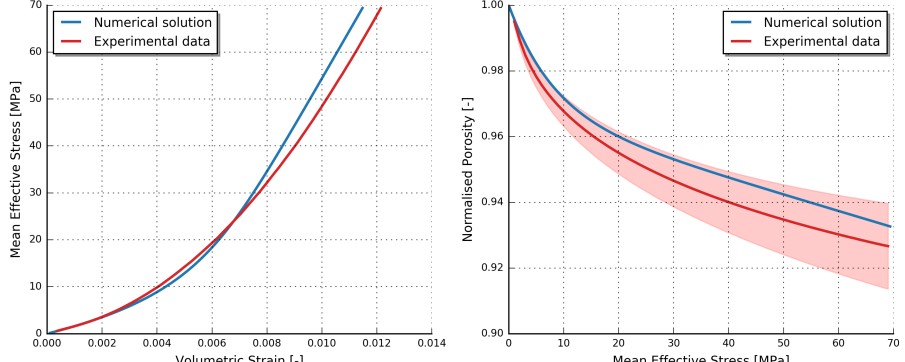

**Figure 2.** Numerical simulations of tri-axial mechanical experiments on a Flechtinger sandstone. Left panel: evolution of the mean effective stress with respect to the volumetric strain. Right panel: evolution of the normalised porosity with respect to the mean effective stress. The non-linear behaviour at low effective stress as observed in the experimental data (red line after Blöcher et al. (2014)) is well captured by the modelling results which take into account a non-linear evolution of the elastic moduli resulting from closure of pre-existing micro-cracks and fissures enhanced by compaction of the porous space followed by stiffening of the rock (increase Hertzian contact surface between solid grains). The modelling results can also capture the compaction-induced reduction of porosity when compared to experimental data (in red) within experimental errors (red shaded area). More details on the theory and results can be found in Jacquey et al. (2015).

where $\rho_f$ is the density of the fluid phase, $n$ is the porosity, $\boldsymbol{v}_f$ the fluid velocity and $Q_f$ is a sink/source term considered null hereafter for the sake of simplicity. In a similar way, for the solid phase we obtain:

$$\frac{\partial\left((1-n)\rho_s\right)}{\partial t}+\nabla\cdot\left((1-n)\rho_s\boldsymbol{v}_s\right)=Q_s \qquad (2)$$

where $\rho_s$ is the density of the solid phase, $\boldsymbol{v}_s$ the solid velocity and $Q_s$ is a sink/source term (considered as null hereafter). We use Darcy's law to describe the conservation of momentum of the fluid phase, which can be expressed in terms of fluid velocity relative to the solid velocity as:

$$\boldsymbol{q}_D=n\left(\boldsymbol{v}_f-\boldsymbol{v}_s\right)=-\frac{\boldsymbol{k}}{\mu_f}\cdot\left(\nabla p_f-\rho_f\boldsymbol{g}\right) \qquad (3)$$

where $\boldsymbol{q}_D$ is a volumetric flow rate per unit of surface area (Darcy velocity), $\boldsymbol{k}$ is the permeability tensor of the porous medium, $\mu_f$ the fluid viscosity and $\boldsymbol{g}$ the gravity vector.

Substituting Eq. 3 into Eq. 1 yields:

$$\frac{\partial\left(n\rho_f\right)}{\partial t}+\nabla\cdot\left(\rho_f\boldsymbol{q}_D\right)+\nabla\cdot\left(n\rho_f\boldsymbol{v}_s\right)=0. \qquad (4)$$





The equations of mass conservations can be rewritten by applying the concept of the Lagrangian (total) derivative with respect to a moving solid, e.g. $\frac{D^s(\bullet)}{Dt} \equiv \frac{\partial(\bullet)}{\partial t} + \nabla(\bullet) \cdot \boldsymbol{v}_s$, and $\frac{D^f(\bullet)}{Dt} \equiv \frac{\partial(\bullet)}{\partial t} + \nabla(\bullet) \cdot \boldsymbol{v}_f$ for a moving fluid. By expanding the fluid mass conservation equation and noting that $\nabla \cdot [(\bullet)\boldsymbol{v}_f] = (\bullet)\nabla \cdot \boldsymbol{v}_f + \nabla(\bullet) \cdot \boldsymbol{v}_f$, Eq. 4 can be rewritten as:

$$\frac{n}{\rho_f} \frac{D^f \rho_f}{Dt} + \frac{D^s n}{Dt} + n\nabla \cdot \boldsymbol{v}_s + \nabla \cdot \boldsymbol{q}_D = 0. \tag{5}$$

In a similar way, it is possible to rework the solid mass balance equation (Eq. 2) to obtain:

$$\frac{(1-n)}{\rho_s} \frac{D^s \rho_s}{Dt} - \frac{D^s n}{Dt} + (1-n)\nabla \cdot \boldsymbol{v}_s = 0. \tag{6}$$

From Eq. 6, it can be noticed that even by considering both the solid skeleton and the pore fluid to be incompressible, the porous rock material will deform (contract or dilate) when fluid is expelled from or injected into the pore space.

Equation 6 can be used to express the evolution of the porosity in term of the Lagrangian derivative with respect to the solid

deformation velocity as:

$$\frac{D^s n}{Dt} = \frac{(1-n)}{\rho_s} \frac{D^s \rho_s}{Dt} + (1-n)\nabla \cdot \boldsymbol{v}_s. \tag{7}$$

Substituting Eq. 7 into Eq. 5, one obtains:

$$\frac{n}{\rho_f} \frac{D^f \rho_f}{Dt} + \frac{(1-n)}{\rho_s} \frac{D^s \rho_s}{Dt} + \nabla \cdot \boldsymbol{v}_s + \nabla \cdot \boldsymbol{q}_D = 0. \tag{8}$$

The first term in the left hand side of Eq. 8 can be expressed in terms of the fluid pore pressure and temperature by thermo-

dynamic differentiation as:

$$\frac{D^f \rho_f}{Dt} = n\left(\frac{1}{K_f} \frac{D^f p_f}{Dt} - \beta_f \frac{D^f T}{Dt}\right) \tag{9}$$

where $\frac{1}{K_f} = \frac{1}{\rho_f}\left(\frac{\partial \rho_f}{\partial p_f}\right)_T$ is the inverse of the fluid bulk modulus and $\beta_f = -\frac{1}{\rho_f}\left(\frac{\partial \rho_f}{\partial T}\right)_{p_f}$ the fluid volumetric thermal expansion coefficient.

The second term in the left hand side of Eq. 8 can also be cast in terms of the problem variables, e.g. pore pressure,

temperature and solid skeleton displacements, by defining a proper constitutive mechanical model. The linear momentum balance equation of the mixture in terms of the effective Cauchy stress tensor $\boldsymbol{\sigma}'(\boldsymbol{x}, t)$ takes the form:

$$\nabla \cdot (\boldsymbol{\sigma}' - \alpha p_f \mathbb{1}) + \rho_b \boldsymbol{g} = 0 \tag{10}$$




where $\mathbb{1}$ is the rank-two identity tensor, $\rho_b$ is the bulk density of the fluid-solid mixture ($\rho_b = n\rho_f + (1-n)\rho_s$) and $\alpha = 1 - \frac{K}{K_s}$ is the Biot coefficient, with $K$ being the drain bulk modulus and $K_s$ the bulk modulus of the solid grains. The geometrical compatibility condition gives the following strain-displacement relation:

$$\boldsymbol{\epsilon} = \frac{1}{2}\left(\nabla\boldsymbol{u} + \nabla^T\boldsymbol{u}\right) = \nabla^s\boldsymbol{u}. \tag{11}$$

Deformation of the solid skeleton is described in terms of thermo-poroelastic response (Biot's consolidation theory) and dissipative plastic behaviour. To simplify the presentation of the constitutive mechanical model, in the following we will consider only small strain conditions, but the theory has also been extended to account for finite deformation. In addition, due to strain history dependence, we detail the formulation in incremental form. Following Biot's theory, (effective) stresses are related to elastic strains via the following relationship:

$$\dot{\boldsymbol{\sigma}}' = \dot{\sigma}'_{ij} = \mathbb{C}_{ijkl}\dot{\epsilon}^e_{kl} = \mathbb{C} : \dot{\boldsymbol{\epsilon}}^e \tag{12}$$

where $\mathbb{C} = \mathbb{C}_{ijkl} = \lambda\delta_{ij}\delta_{kl} + 2G\delta_{ik}\delta_{jl}$ is the rank-four elastic stiffness tensor, with $\lambda$ and G being the first and second (shear) Lamé moduli respectively.

    The stress-strain constitutive relation given by Eq. 12, can then be used to find an expression for the material derivative of the solid density in terms of the problem variables (second term in Eq. 8) as:

$$\frac{(1-n)}{\rho_s}\frac{D^s\rho_s}{Dt} = \frac{(\alpha-n)}{K_s}\frac{D^sp_f}{Dt} - (1-n)\beta_s\frac{D^sT}{Dt} - \frac{1}{K_s}\frac{D^s\bar{\sigma}'}{Dt} \tag{13}$$

where $\bar{\sigma}'$ indicates the mean effective stress and $\beta_s$ the volumetric thermal expansion coefficient of the solid grains.

    Substituting Eq. 9 and Eq. 13 into Eq. 8, we obtain:

$$\frac{n}{K_f}\frac{D^fp_f}{Dt} - n\beta_f\frac{D^fT}{Dt} + \frac{(\alpha-n)}{K_s}\frac{D^sp_f}{Dt}(1-n)\beta_s\frac{D^sT}{Dt} - \frac{1}{K_s}\frac{D^s\bar{\sigma}'}{Dt} + \dot{\epsilon}_{kk} + \nabla\cdot\boldsymbol{q}_D = 0 \tag{14}$$

where we have expressed the gradient of the solid deformation velocity in terms of the volumetric component of the total stain
rate tensor, $\nabla\cdot\boldsymbol{v}_s = \nabla\cdot\dot{\boldsymbol{u}} = \dot{\epsilon}_{kk}$.

    By making use of the definition of the total derivative and given the stress-strain constitutive equation (Eq. 12), Eq. 14 can be recast as:

$$\frac{1}{M_b}\frac{\partial p_f}{\partial t} - \beta_b\frac{\partial T}{\partial t} - (1-\alpha)\dot{\epsilon}^e_{kk} + \dot{\epsilon}_{kk} + \nabla\cdot\boldsymbol{q}_D + \frac{n}{K_f}\nabla p_f\cdot\boldsymbol{v}_f + \frac{(\alpha-n)}{K_s}\nabla p_f\cdot\boldsymbol{v}_s\left(n\beta_f\nabla T\cdot\boldsymbol{v}_f + (1-n)\beta_s\nabla T\cdot\boldsymbol{v}_s\right) - \frac{1}{K_s}\nabla\bar{\sigma}'\cdot\boldsymbol{v}_s = 0 \tag{15}$$

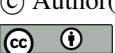



where $\frac{1}{M_b} = \frac{n}{K_f} + \frac{(\alpha - n)}{K_s}$ is the specific storage of the porous medium (reciprocal of the Biot modulus $M_b$) and $\beta_b = n\beta_f + (1 - n)\beta_s$ is the bulk volumetric thermal expansion coefficient.

It is possible to rewrite Eq. 15 in terms of the solid velocity only, by integrating the momentum balance equation (Eq. 3) as:

$$\frac{1}{M_b}\frac{\partial p_f}{\partial t} - \beta_b \frac{\partial T}{\partial t} - (1-\alpha)\dot{\epsilon}^e_{kk} + \dot{\epsilon}_{kk} + \nabla \cdot \boldsymbol{q}_D + \boldsymbol{q}_D \cdot \left( \frac{1}{K_f}\nabla p_f - \beta_f \nabla T \right) + \boldsymbol{v}_s \cdot \left( \frac{1}{M_b}\nabla p_f - \beta_b \nabla T - \frac{1}{K_s}\nabla \bar{\sigma}' \right) = 0 \quad (16)$$

where the last two terms can be considered as second order correction terms taking into account non-linear advective effects.

To quantify any permanent (irreversible) deformation of the material due to inelastic processes we make use of the concept of eigenstrain ($\epsilon^*$) as derived from micromechanics (Mura, 1987). By assuming small strain approximation, the total strain of the material $\epsilon$ can be decomposed as the sum of the elastic strain ($\epsilon^e$) and eigenstrain components ($\epsilon^*$) as:

$$\epsilon = \epsilon^e + \epsilon^*. \quad (17)$$

Therefore, Eq. 16 can be written as:

$$\frac{1}{M_b}\frac{\partial p_f}{\partial t} - \beta_b \frac{\partial T}{\partial t} + \alpha\dot{\epsilon}_{kk} + (1-\alpha)\dot{\epsilon}^*_{kk} + \nabla \cdot \boldsymbol{q}_D + \boldsymbol{q}_D \cdot \left( \frac{1}{K_f}\nabla p_f - \beta_f \nabla T \right) + \boldsymbol{v}_s \cdot \left( \frac{1}{M_b}\nabla p_f - \beta_b \nabla T - \frac{1}{K_s}\nabla \bar{\sigma}' \right) = 0 \quad (18)$$

In the present study we focus on two major kinds of residual deformation, that is thermal expansion and plastic flow, i.e. $\epsilon^* = \epsilon^*_{ij} = \epsilon^{*T}_{ij} + \epsilon^{*p}_{ij} = \epsilon^{*T} + \epsilon^{*p}$, though additional processes including for example swelling, fatigue or phase transformations can be relatively easily integrated in the current formulation.

Thermal strains are related to deformation induced by temperature changes inside the material, and can be therefore expressed by:

$$\dot{\epsilon}^{*T} = \frac{1}{3}\beta_b \dot{T} \mathbb{1} \quad (19)$$

where $\dot{T}$ is the relative temperature rate.

We determine the plastic component of the strain tensor (Eq. 17) by making use of the normality rule as:

$$\dot{\epsilon}^{*p} = \dot{\gamma}\frac{\partial \mathcal{Q}}{\partial \boldsymbol{\sigma}'} \quad (20)$$

where $\dot{\gamma} = \dot{\gamma}(\boldsymbol{\sigma}', \kappa)$ is the plastic multiplier satisfying the classical Kuhn-Tucker conditions ($\dot{\gamma} \geq 0, \mathcal{F} \leq 0, \dot{\gamma}\mathcal{F} = 0$) with $\mathcal{F}(\boldsymbol{\sigma}', \kappa)$ being the yield surface. $\mathcal{Q} = \mathcal{Q}(\boldsymbol{\sigma}', \kappa)$ is the plastic potential function giving the direction of the plastic strain increment, and $\kappa$ represents the vector of internal variables affecting the evolution of the yield surface during loading and unloading of the material.



Therefore, Eq. 18 can be finally written for thermal and plastic eigenstrains as:

$$\frac{1}{M_b}\frac{\partial p_f}{\partial t} - \alpha\beta_b\frac{\partial T}{\partial t} + \alpha\dot{\epsilon}_{kk} + (1-\alpha)\dot{\epsilon}_{kk}^{*p} + \nabla\cdot\boldsymbol{q}_D + \boldsymbol{q}_D\cdot\left(\frac{1}{K_f}\nabla p_f - \beta_f\nabla T\right) + \boldsymbol{v}_s\cdot\left(\frac{1}{M_b}\nabla p_f - \beta_b\nabla T - \frac{1}{K_s}\nabla\bar{\sigma}'\right) = 0. \quad (21)$$

Similarly, the evolution of porosity can therefore be expressed using Eqs. 7 and 13 as well as the strain decomposition as:

$$\frac{\partial n}{\partial t} = \frac{(\alpha-n)}{K_s}\frac{\partial p_f}{\partial t} - \beta_n\frac{\partial T}{\partial t} + (\alpha-n)\dot{\epsilon}_{kk} + (1-\alpha)\dot{\epsilon}_{kk}^{*p} + \boldsymbol{v}_s\cdot\left(\frac{(\alpha-n)}{K_s}\nabla p_f - (1-n)\beta_s\nabla T - \frac{1}{K_s}\nabla\bar{\sigma}' - \nabla n\right) \quad (22)$$

where $\beta_n = (1-n)\beta_s - (1-\alpha)\beta_b$ is the volumetric thermal expansion coefficient of the pores and the last term can be considered as being of second order.

The balance of energy for the fluid-rock mixture, assuming local thermal equilibrium between the two phases, and neglecting the dissipation of mechanical energy due to deformation of the solid phase reads as:

$$T\frac{\partial(\rho c)_b}{\partial t} + (\rho c)_b\frac{\partial T}{\partial t} + \nabla\cdot(\rho_f c_f\boldsymbol{q}_D T - \lambda_b\nabla T) - \dot{H} = 0 \quad (23)$$

where $(\rho c)_b = n\rho_f c_f + (1-n)\rho_s c_s$ is the bulk specific heat of the porous medium, $\lambda_b = n\lambda_f + (1-n)\lambda_s$ is the bulk thermal conductivity, and $\dot{H}$ is a rate of energy production. The first term of the left hand side of Eq. 23 takes into account secondary, non-Boussinesq dissipative effects related to the pressure temperature dependency of the bulk storage, which are usually rewritten only considering variable fluid and solid density as:

$$T\frac{\partial(\rho c)_b}{\partial t} = (1-n)c_s T\left(\frac{\rho_s}{K_s}\frac{\partial p_f}{\partial t} - \rho_s\beta_s\frac{\partial T}{\partial t}\right) + nc_f T\left(\frac{\rho_f}{K_f}\frac{\partial p_f}{\partial t} - \rho_f\beta_f\frac{\partial T}{\partial t}\right) \quad (24)$$

The use of a conservative finite element formulation in Eq. 23 ensures the balance of the fluid enthalpy ($\rho_f c_f T$) both at element and node levels. Therefore, it guarantees that we avoid accumulating in time unbalances in intercell fluxes as it is usually the case when relying on non-conservative formulations for convective-type problems (Giudice et al., 1992).

The energy conservation equation as given by Eq. 23 is valid for a porous medium in the absence of any thermoelastic coupling. Following Biot consolidation theory, it is possible to consider the effects of the solid elastic deformation on the 20 temperature distribution by augmenting Eq. 23 with a thermoelastic dissipation rate term, as:

$$T\frac{\partial(\rho c)_b}{\partial t} + (\rho c)_b\frac{\partial T}{\partial t} + T_0\beta_b\dot{\epsilon}_{kk}^e + \nabla\cdot(\rho_f c_f\boldsymbol{q}_D T - \lambda_b\nabla T) - \dot{H} = 0 \quad (25)$$

where $T_0$ represents the absolute temperature of the porous medium in a stress-free state. Equation 25 can be easily modified to take into account additional thermal effects from fluid dilation and shear heating stresses.



## 3   Numerical implementation

In order to solve the system of coupled and non-linear equations as described above (Eq. 10, Eq. 21, and Eq. 25), a number of interconnected issues must be taken into account. These include for example the choice of the spatial discretisation adopted, the time stepping scheme and temporal integration, iterative solvers and preconditioners. In what follows, we describe the methods
adopted in GOLEM in order to tackle these issues. In the next chapter, we also present some numerical applications that serve as code benchmarking and illustrate the capability of the simulator to solve for problems of different degree of difficulty at optimal cost.

GOLEM is an open source simulator specifically developed for Thermo-Hydro-Mechanical coupled applications in fractured geological systems, supporting 1D, 2D, and 3D computations in a single code implementation. It builds on the object-oriented
numerical framework MOOSE developed at the Idaho National Laboratory (Gaston et al., 2009). MOOSE provides a flexible, massive parallel (including both MPI and multi-threading) platform to solve for multiphysics and multi-component problems in an implicit manner. It relies on state of the art and extensively tested libraries developed both at universities and national laboratories, as the libMesh library (Kirk et al., 2006) developed at the University of Austin in Texas for capabilities related to parallel finite element method, and a suite of scalable parallel nonlinear and linear solvers (PETSc, Balay et al. (2016);
Trilinos, Heroux et al. (2005); and Hypre, Chow et al. (1998)). The use of different open-source libraries provides a modular structure to the framework, which allows the developer of an application to only concentrate on the high level description of the multiphysics problem and to maintain the code relatively compact. Following the basic structure of the MOOSE framework, GOLEM has also been developed as a modular application, where each module is responsible for the solution of a specific physical process. This aspect makes easy any further modification, adjustment and improvement of the program with limited
efforts from the user's side. At the present stage, GOLEM is available by contacting one of the two authors and it comes together with a suite of relatively simple benchmark problems. Input files for running all applications presented in the manuscript can also be obtained upon request.

### 3.1   Variational formulation and its numerical solution

The finite element discretisation is based on the weak form, in an integral sense, of the system of partial differential equations
as derived in the previous paragraph. At this purpose, we consider the porous matrix to be described by a close domain of volume $\Omega \subset \Re^n$ bounded by a boundary $\Gamma \subset \Re^{(n-1)}$. Given the length to width ratio typical of fractures, a discrete fracture is





represented by a lower dimensional element of volume $\Omega_f \subset \Re^{(n-1)}$ and surface area $\Gamma_f \subset \Re^{(n-2)}$. The corresponding weak form of the governing equations are then derived by applying the method of the weighted residuals as:

$$
\int_\Omega \omega \frac{1}{M_b} \frac{\partial p_f}{\partial t} d\Omega - \int_\Omega \omega \alpha \beta_b \frac{\partial T}{\partial t} d\Omega
$$

$$
+ \int_\Omega \omega \left( \alpha \dot{\epsilon}_{kk} + (1-\alpha) \dot{\epsilon}_{kk}^{*p} \right) d\Omega
$$

$$
- \int_\Omega \nabla \boldsymbol{\omega} \cdot \boldsymbol{q}_D d\Omega + \int_{\Gamma_{q_H}} \omega \left( \boldsymbol{q}_D \cdot \boldsymbol{n}_{\Gamma_{q_H}} \right) d\Gamma
$$

$$
+ \int_\Omega \omega \boldsymbol{q}_D \cdot \left( \frac{1}{K_f} \nabla p_f - \beta_f \nabla T \right) d\Omega
$$

$$
+ \int_\Omega \omega \boldsymbol{v}_s \cdot \left( \frac{1}{M_b} \nabla p_f - \beta_b \nabla T - \frac{1}{K_s} \nabla \bar{\sigma}' \right) d\Omega = 0 \tag{26}
$$

$$
\int_\Omega \omega (\rho c)_b \frac{\partial T}{\partial t} d\Omega - \int_\Omega \nabla \omega \cdot \left( (\rho c)_f \boldsymbol{q}_D T - \lambda_b \nabla T \right) d\Omega + \int_{\Gamma_{q_T}} \omega \left( (\rho c)_f \boldsymbol{q}_D T - \lambda_b \nabla T \right) \cdot \boldsymbol{n}_{\Gamma_{q_T}} d\Gamma + \int_\Omega \omega \dot{H} d\Omega = 0 \tag{27}
$$

$\quad \int_\Omega \nabla^s \boldsymbol{\omega} : (\boldsymbol{\sigma}' - \alpha p_f \mathbb{1}) d\Omega - \int_\Omega \boldsymbol{\omega} \cdot \rho_b \boldsymbol{g} d\Omega - \int_\Gamma \boldsymbol{\omega} \cdot (\boldsymbol{\sigma}' - \alpha p_f \mathbb{1}) \cdot \boldsymbol{n}_{\Gamma_{q_M}} d\Gamma = 0 \tag{28}$

where in Eq. 26, Eq. 27, and Eq. 28 we have omitted all supra-(sub)scripting for easiness in the notation.

The equations derived above describe an initial and boundary value problem, for which proper boundary and initial conditions need to be assigned. These can be set by either prescribing the value of the problem variables along or their flux across a portion of the boundary. More precisely, the model discussed in the previous paragraph, must satisfy the following set of

10 boundary conditions:

- Prescribed displacement, pore pressure and temperature, i.e. $\boldsymbol{u} = \bar{\boldsymbol{u}}$, $p_f = \bar{p}_f$, $T = \bar{T}$ on $\Gamma_u$, $\Gamma_{p_f}$, $\Gamma_T$ respectively.

- Equilibrium of boundary stresses and external loads (last two terms in Eq. 28).

- Continuity of the fluid flux across the imposed boundary, i.e. $\boldsymbol{q}_D \cdot \boldsymbol{n}_{\Gamma_{q_H}} = \bar{q}_H$ on $\partial \Omega_{q_H}$ where $\bar{q}_H$ is the rate of in/out-flow per unit of area across the boundary surface.

- Continuity of the total (diffusive plus advective) heat flow, i.e. $\left( (\rho c)_f \boldsymbol{q}_D T - \lambda_b \nabla T \right) \cdot \boldsymbol{n}_{\Gamma_{q_T}} = \bar{q}_T$ on $\Gamma_{q_T}$ where $\bar{q}_T$ is the rate of in/out-heat flow per unit of area across the boundary surface.

The resulting system of equations together with a proper choice of boundary and initial conditions is discretised spatially by the Finite Element Method, while the temporal discretisation is done by traditional Finite Difference techniques. The nodal





values of the primary variables of the problem, temperature ($T^n$), pore pressure ($p_f^n$), and the deformation vector of the solid skeleton ($\boldsymbol{u}^n$) are approximated by linear Lagrangian interpolation polynomial functions as:

$$
\begin{aligned}
T^n &= \sum_{i=1}^{i=N_T} T_i^n w_i^T(\boldsymbol{x}) \\
p_f^n &= \sum_{i=1}^{i=N_{p_f}} p_{f\,i}^{\,n} w_i^{p_f}(\boldsymbol{x}) \\
\boldsymbol{u}^n &= \sum_{i=1}^{i=N_u} u_i^n \boldsymbol{w}_i^u(\boldsymbol{x})
\end{aligned}
\tag{29}
$$

Though higher order polynomials are also available through the libMesh library, we rely on linear finite element approximation for all variables. Indeed, we have found that, for the kinds of problems as those that will be presented in the next paragraph, higher order approximations do not necessarily guarantee a better convergence of the solution, being subjected at the same time to severe under-(over)shooting numerical effects in the presence of sharp gradients.

Single-nodded, zero thickness finite elements are used to explicitly represent individual fractures, the latter assumed to be clean and fully saturated (i.e. $n = 1$ and $M_b = K_f$). It follows that fractures can be parametrised as based on the concept of their effective aperture, which provides a quantitative measure of the geometric width of the fracture surface. The use of single-nodded finite elements impose continuity of gradients in both pore pressure and temperature across the fracture width, as well as the absence of any shear and normal strain acting on the fracture plane. While the latter assumption simplifies the problem formulation, it prevents to include the effects of local fracture mechanics in the current formulation. Therefore, fractures are only considered as having a distinct hydraulic and thermal behaviour with respect to the porous domain. To extend the current formulation to include mechanical feedback effects from localised deformation, nucleation and fracture propagation is part of ongoing activities.

The mass balance equation for a discrete fracture then reads as:

$$
\int_{\Omega_f}\int_{-\frac{b}{2}}^{\frac{b}{2}} \omega \frac{1}{K_f} \frac{\partial p_f}{\partial t} dz' d\Omega_f - \int_{\Omega_f}\int_{-\frac{b}{2}}^{\frac{b}{2}} \bar{\nabla}\omega \cdot \boldsymbol{q}_D dz' d\Omega_f + \int_{\Gamma_f}\int_{-\frac{b}{2}}^{\frac{b}{2}} \omega \boldsymbol{q}_D \cdot \hat{n} dz' d\Gamma + \Lambda_H^+ + \Lambda_H^- = 0
\tag{30}
$$

where $\Lambda_H^{\pm}$ are leakage terms (weak form) across each of the two sides of the fracture surface into the surrounding porous domain and $b$ is the aperture of the fracture.

In a similar manner, the energy balance equation for a fracture element reads as:

$$
\int_{\Omega_f}\int_{-\frac{b}{2}}^{\frac{b}{2}} \omega \left( c_f \rho_f \frac{\partial T}{\partial t} + H \right) dz' d\Omega_f - \int_{\Omega_f}\int_{-\frac{b}{2}}^{\frac{b}{2}} \nabla_f^T \omega \cdot (c_f \rho_f \boldsymbol{q}_D T - \lambda_f \nabla_f T) dz' d\Omega_f + \int_{\partial\Omega}\int_{-\frac{b}{2}}^{\frac{b}{2}} \omega \left( c_f \rho_f \boldsymbol{q}_D T - \lambda_f \nabla_f T \right) \cdot \hat{n} dz' d\Omega_f + \Lambda_T^+ + \Lambda_T^- = 0
$$





$$\tag{31}$$

where $\Lambda_T^{\pm}$ quantify the amount of heat leaking from the fracture surfaces into the porous medium domain.

In equations 30 and 31, integration is done in local coordinates of the fracture tangential and normal directions respectively $(x', y', z')$. This enables to superimpose the discretised conservative equations for the porous medium and the fractures at the nodal location shared by the two elements, where the fluid mass and heat fluxes also cancelled out. Therefore the weak form of the conservation equations can be simplified as:

$$\int_{\Omega_f} b\omega \frac{1}{K_f} \frac{\partial p_f}{\partial t} d\Omega_f - \int_{\Omega_f} b\nabla_f \omega^T \cdot \boldsymbol{q}_D d\Omega_f + \int_{\partial\Omega_f} b\omega \boldsymbol{q}_D \cdot \hat{\boldsymbol{n}} dS_f = 0 \tag{32}$$

and

$$\int_{\Omega_f} b\omega(c_f\rho_f \frac{\partial T}{\partial t} + H)d\Omega_f - \int_{\Omega_f} b\nabla_f^T \omega \cdot (c_f\rho_f \boldsymbol{q}_D T - \lambda_f \nabla_f T)d\Omega_f \quad + \int_{\partial\Omega_f} b\omega(c_f\rho_f \boldsymbol{q}_D T - \lambda_f \nabla_f T) \cdot \hat{\boldsymbol{n}} dS_f = 0 \tag{33}$$

where we have made no distinction between the effective hydraulic aperture and the mechanical aperture ($b_h = b_m = b = \int_{-\frac{b}{2}}^{\frac{b}{2}} dz'$).

Derivatives of the test functions and of direction-dependent material properties with respect to the system of global coordinates are computed by standard coordinate transformation, i.e. $\frac{\partial N_i}{\partial x_i} = J_{ij}^{-1} \frac{\partial N_j}{\partial x_j'}$ where $J_{ij}$ is the Jacobian matrix of the mapping between global ($x_i$) and local ($x_i'$) coordinates. Transformation in local coordinates for lower dimensional elements is achieved by computing the rotational matrix, $\boldsymbol{R} = \begin{bmatrix} \cos(x',x) & \cos(x',y) & \cos(x',y) \\ \cos(y',x) & \cos(y',y) & \cos(y',y) \\ \cos(z',x) & \cos(z',y) & \cos(z',y) \end{bmatrix}$ with $\cos(x_i',x_i)$ being the directional cosines.

Coordinate transformation is applied to all direction-dependent (i.e. tensorial) material properties as well as to the directional derivatives as:

$$\boldsymbol{N} = \boldsymbol{R}\boldsymbol{I}_f \boldsymbol{N}' \boldsymbol{I}_f^T \boldsymbol{R}^T \tag{34}$$

where $\boldsymbol{N}$ and $\boldsymbol{N}'$ are direction-dependent material properties in global and local coordinates respectively, and $\boldsymbol{I}_f$ is the unit tangent vector of local coordinates.

The above system of coupled equation can be rewritten in a more concise form as:

$$\bar{\boldsymbol{S}}\frac{d\bar{\boldsymbol{x}}}{dt} + \bar{\boldsymbol{K}}\bar{\boldsymbol{x}} - \bar{\boldsymbol{F}} = \boldsymbol{R}(\hat{\boldsymbol{u}}) \tag{35}$$

where $\bar{\boldsymbol{S}}$ is the nodal coefficient storage matrix, $\bar{\boldsymbol{K}}$ denotes the stiffness matrix of the problem, $\bar{\boldsymbol{F}}$ is the load vector, $\bar{\boldsymbol{x}}$ is the vector of the problems variables, and $\boldsymbol{R}(\hat{\boldsymbol{u}})$ is the residual from the discrete approximation. We make use of an unconditionally





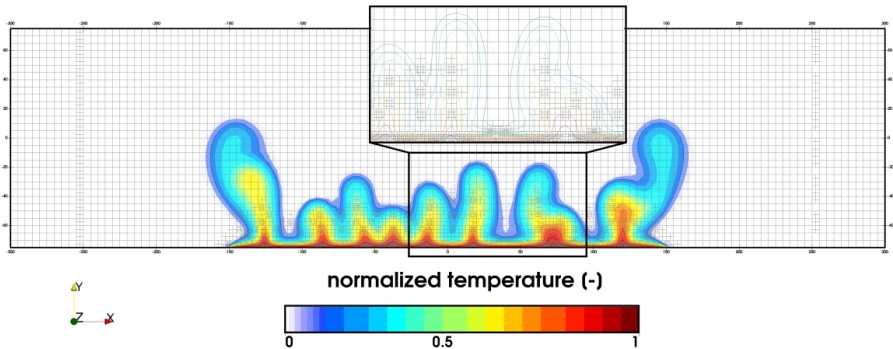

**Figure 3.** Example of a non-linear convective flow problem after Elder (1967). An homogeneous, isotropic and saturated porous medium is heated from below with fluid density and viscosity being a function of temperature and pressure (IAPWS, 2008a, b). Due to the temporal evolution in fluid property gradients, thermal buoyant forces develop thus giving rise to an unstable convective flow regime as illustrated by the temperature isolines. H-mesh adaptive refinement with local error estimate based on the L-2 norm projection of the intercell thermal flux vectors and a two-refinement cycle per time step has been enforced in order to guarantee numerical stability even in the presence of sharp thermal gradients (see close up view).

stable, backward Euler method to integrate Eq. 35 in time, and to arrive at the monolithic form of the coupled system. The monolithic form of Eq. 35 is then solved iteratively by classical Newton's method, where different linear solvers (e.g. GM-RES and its flexible variant, FGMRES) and preconditioners (including also Newton-Krylov-Schwartz domain decomposition techniques) are available from the open source algebraic libraries.

## 3.2 Nonlinearities and their stabilisation

The coefficient matrices in Eq. 35 contain off-diagonal components, thus giving rise to a highly-nonlinear problem to be solved for. Non-linearities arise due the non-linear constitutive relationship between problem variables and material properties, as well as to the presence of internal feedback effects among the different processes. The constitutive relation linking structure, properties and primary variables considered include equation of states for fluid density, thermal compressibility and expansivity and fluid viscosity with respect to the $p_f - T$ state of the system (IAPWS, 2008a, b), as well as porosity and permeability relations as a function of variation in the thermal, mechanical an hydraulic state of the system, see Fig. 3 and Fig. 2 respectively.

For problems involving heat transport, if the convective term dominates over the diffusive term, the solution of the energy transport equation typically contains interior and boundary layers and its solution by the Galerkin finite element technique is usually globally unstable (Nield and Bejan, 2012). Numerical instabilities which takes the form of spurious oscillations around sharp gradients and boundary layers and likely lead to non convergent solutions, have been observed to occur independent of the level of mesh refinement (Diersch and Kolditz, 1998). In such situations, in order to enhance the stability and accuracy of the finite element solution, some sort of stabilisation has to be added to the discrete formulation. In GOLEM, we have opted for the Streamline Upwind Petro/Galerkin (SUPG) method, which can be conceived as adding an upwind perturbation along





computed streamlines to the standard Galerkin formulation (Brooks and Hughes, 1982). The SUPG discretisation of Eq. 27 is obtained by making use of an 'enriched' weighting functions, $\omega^*$, as:

$$\omega^* = \omega + p \tag{36}$$

where $\omega$ is the continuous weighting function and $p = \tau \nabla \omega \cdot \nabla q_D \|q_D\|$ is a discontinuous streamline upwind correction. Note that by its definition, $\omega^*$ is no longer continuous across inter-element boundaries. A critical question is related to the choice of the upwind parameter $\tau$, which might influence the stability and accuracy of the discrete solution. In all simulation presented in the manuscript, when needed, we have made used of the formula as presented in (Galeão et al., 2004):

$$\tau = \frac{h}{2\|q_D\|p_p}\left(\coth(P_e) - \frac{1}{P_e}\right) \tag{37}$$

where $h$ is the diameter of the finite element adopted along the direction of $q_D$, $p_p$ is the order of approximation considered in the interpolation, $\|\cdot\|$ is the Euclidean norm, and $P_e = \frac{\|q_D\|h}{2kp}$ is the local Peclet number. A major disadvantage of the SUPG method is to add numerical diffusion to all elements characterised by a high Peclet number, even in the absence of sharp thermal gradients. We leave as a subject of future studies the implementation of more recent and more sophisticated damping methods, e.g. entropy viscosity method as described for instance in Guermond et al. (2011).

### 3.3 Plasticity and return-map algorithm

We consider plastic deformation in terms of a Drucker-Prager frictional plastic yield criterion (Drucker, 1950) in which the onset of yield is a function of the first and second invariants of the effective stress tensor $J_1 = \sigma'_{kk}, \quad J_2 = \frac{1}{2}\left(\tau_{ij}\tau_{ij}\right)$ respectively where $\tau_{ij}$ is the deviatoric part of the effective stress tensor ($\tau_{ij} = \sigma'_{ij} - \frac{1}{3}J_1\delta_{ij}$):

$$\mathcal{F} = \sqrt{J_2 + \epsilon_0^2} + \frac{\sin(\varphi)}{3}J_1 - C\cos(\varphi) \tag{38}$$

where $\varphi$ and $C$ are the Mohr-Coulomb friction angle and cohesion respectively and $\epsilon_0$ is a small non-hardening parameters here introduced to relax the singularity at the cone's tip of the Drucker-Prager yield envelope. An important aspect relates to the form of plastic potential function adopted. It has been shown that associated flow rule might lead to overestimate the dilation of the rocks and it generally results in a overly weaker response of the rock to loading (Vermeer and de Borst, 1984; Jiang and Xie, 2011). We avoid these issues by implementing a non-associated form of Drucker-Prager, in which the plastic potential function, $\mathcal{G}$, is considered to depend on the dilation angle, $\psi$ as:

$$\mathcal{G} = \sqrt{J_2} + \frac{\sin(\psi)}{3}J_1 \tag{39}$$





and, in consequence, the unnormalised flow directions can be derived as:

$$\frac{\partial \mathcal{G}}{\partial \boldsymbol{\sigma}'_{ij}} = \frac{1}{2\sqrt{J_2}} \frac{\partial J_2}{\partial \boldsymbol{\sigma}'_{ij}} + \frac{\sin(\psi)}{3} \delta_{ij} \tag{40}$$

Note that in our formulation we take also into account possible degradation of the strength of the rock subjected to loading in terms of hardening (and softening) of the internal parameters (friction angle, cohesion and dilation angle) as a function of

the accumulated plastic strain (internal variable $\boldsymbol{\kappa}$).

The stress update procedure is conducted via a return-map algorithm based on the closest point projection on the yield surface (Simo and Hughes, 1998) within a Newton-Raphson procedure. This algorithm is presented in incremental form in the following. Subscript $n$ refers to a value at the time $t_n$, that is $\sigma_n = \sigma(t_n)$ and superscript $(k)$ refers to the $k^{th}$ iteration in the Newton-Raphson procedure. We use the following notation for sake of simplicity: $\partial_{\boldsymbol{\sigma}'} \cdot = \frac{\partial \cdot}{\partial \boldsymbol{\sigma}'}$.

1. If plastic loading ($\mathcal{F}_{n+1}^{trial} > 0$), then the increment of plastic multiplier is also positive according to the Kuhn-Tucker conditions, $\Delta\gamma > 0$. Define the system of equations with the residuals to minimise, the plastic flow residual $\boldsymbol{R}_{\boldsymbol{\epsilon},n+1}$, the internal parameter residuals $\boldsymbol{R}_{\boldsymbol{\kappa},n+1}$ and the yield condition for this time step $\mathcal{F}_{n+1}$ as:

$$\begin{cases} \boldsymbol{R}_{\boldsymbol{\epsilon},n+1} &= -\boldsymbol{\epsilon}_{n+1}^{*p} + \boldsymbol{\epsilon}_n^{*p} + \Delta\gamma \partial_{\boldsymbol{\sigma}'} \mathcal{G} = -\Delta\boldsymbol{\epsilon}_{n+1}^{*p} + \Delta\gamma \partial_{\boldsymbol{\sigma}'} \mathcal{G} \\ \boldsymbol{R}_{\boldsymbol{\kappa},n+1} &= \boldsymbol{\kappa}_{n+1} + \boldsymbol{\kappa}_n - \Delta\gamma \mathcal{H} = \Delta\boldsymbol{\kappa}_{n+1} - \Delta\gamma \mathcal{H} \\ \mathcal{F}_{n+1} &= \mathcal{F}\left(\boldsymbol{\sigma}'_{n+1}, \boldsymbol{\kappa}_{n+1}\right) \end{cases}$$

where $\mathcal{H}$ is the hardening flow rule for the internal parameter $\boldsymbol{\kappa}$, that is $\dot{\boldsymbol{\kappa}} = -\dot{\gamma}\mathcal{H}$.

2. This system of equation is then linearised as follow:

$$\begin{cases} \boldsymbol{R}_{\boldsymbol{\epsilon},n+1}^{(k)} + \partial_{\boldsymbol{\sigma}'} \boldsymbol{R}_{\boldsymbol{\epsilon},n+1}^{(k)} : \Delta\boldsymbol{\sigma}'^{(k)}_{n+1} + \partial_{\boldsymbol{\kappa}} \boldsymbol{R}_{\boldsymbol{\epsilon},n+1}^{(k)} : \Delta\boldsymbol{\kappa}_{n+1}^{(k)} &= 0 \\ \boldsymbol{R}_{\boldsymbol{\kappa},n+1}^{(k)} + \partial_{\boldsymbol{\sigma}'} \boldsymbol{R}_{\boldsymbol{\kappa},n+1}^{(k)} : \Delta\boldsymbol{\sigma}'^{(k)}_{n+1} + \partial_{\boldsymbol{\kappa}} \boldsymbol{R}_{\boldsymbol{\kappa},n+1}^{(k)} : \Delta\boldsymbol{\kappa}_{n+1}^{(k)} &= 0 \\ \mathcal{F}_{n+1}^{(k)} + \partial_{\boldsymbol{\sigma}'} \mathcal{F}_{n+1}^{(k)} : \Delta\boldsymbol{\sigma}'^{(k)}_{n+1} + \partial_{\boldsymbol{\kappa}} \mathcal{F}_{n+1}^{(k)} : \Delta\boldsymbol{\kappa}_{n+1}^{(k)} &= 0 \end{cases}$$

which can be expressed as the following matrix system:

$$\boldsymbol{J}\boldsymbol{x} = \boldsymbol{R}$$

where $\boldsymbol{J}$ is the jacobian matrix:

$$\boldsymbol{J} = \begin{bmatrix} \left(\mathbb{C}^{-1} + \Delta\gamma_{n+1}^{(k)} \partial^2_{\boldsymbol{\sigma}'\boldsymbol{\sigma}'} \mathcal{G}_{n+1}^k\right) & \Delta\gamma_{n+1}^{(k)} \partial^2_{\boldsymbol{\sigma}'\boldsymbol{\kappa}} \mathcal{G}_{n+1}(k) & \partial_{\boldsymbol{\sigma}'} \mathcal{G}_{n+1}^{(k)} \\ \Delta\gamma_{n+1}^{(k)} \partial_{\boldsymbol{\sigma}'} \mathcal{H}_{n+1}^{(k)} & \mathbb{1} + \Delta\gamma_{n+1}^{(k)} \partial_{\boldsymbol{\kappa}} \mathcal{H}_{n+1}^{(k)} & \mathcal{H}_{n+1}^{(k)} \\ \partial_{\boldsymbol{\sigma}'} \mathcal{F}_{n+1}^{(k)} & \partial_{\boldsymbol{\kappa}} \mathcal{F}_{n+1}^{(k)} & 0 \end{bmatrix}$$





$x$ is the vector of unknowns to compute:

$$x = \begin{bmatrix} \Delta\boldsymbol{\sigma}_{n+1}^{\prime(k)} \\ \Delta\boldsymbol{\kappa}_{n+1}^{(k)} \\ \Delta^2\gamma_{n+1}^{(k)} \end{bmatrix}$$

and $\boldsymbol{R}$ is the residuals vector:

$$\boldsymbol{R} = \begin{bmatrix} -\boldsymbol{R}_{\epsilon,n+1}^{(k)} \\ -\boldsymbol{R}_{\kappa,n+1}^{(k)} \\ -\mathcal{F}_{n+1}^{(k)} \end{bmatrix}.$$

3. The aforementioned matrix system of equation is solved at each $k^{th}$ iteration using routines from the PETSc library (Balay et al., 2016) for the increment of stress, internal parameter and plastic multiplier ($\Delta\boldsymbol{\sigma}_{n+1}^{\prime(k)}$, $\Delta\boldsymbol{\kappa}_{n+1}^{(k)}$ and $\Delta^2\gamma_{n+1}^{(k)}$).

4. The variables are updated at the end of the $k^{th}$ iteration:

$$\boldsymbol{\sigma}_{n+1}^{\prime(k+1)} = \boldsymbol{\sigma}_{n+1}^{\prime(k)} + \Delta\boldsymbol{\sigma}_{n+1}^{\prime(k)}$$
$$\boldsymbol{\epsilon}_{n+1}^{*p(k+1)} = \boldsymbol{\epsilon}_{n+1}^{*p(k)} - \mathbb{C}^{-1} : \Delta\boldsymbol{\sigma}_{n+1}^{\prime(k)}$$
$$\Delta\gamma_{n+1}^{(k+1)} = \Delta\gamma_{n+1}^{(k)} + \Delta^2\gamma_{n+1}^{(k)}$$

5. Steps 1 to 4 are repeated until the residuals reached minimum threshold values.

We have tested and validated the above described return-map algorithm to update the elasto-plastic deformation against available algorithms in the MOOSE tensor mechanics module.

## 4 Results

In this section we present five different applications of the numerical simulator. These applications are intended to test the ability of the simulator to deal both with single processes and their coupling. By starting with simplistic benchmarks, for which analytical solutions exist, we gradually increase the complexity of the problem formulation in order to demonstrate the applicability of the approach to realistic operational cases. An application to an actual field study case based on an injection test performed at the Groß Schönebeck geothermal site (North Germany) is the subject of a separate publication (Jacquey et al. (under review)).





**Table 1.** Fluid properties for the example of heat transport in a fracture

| Property name | Symbol | Value | SI unit |
|---|---|---|---|
| Fluid density | $\rho_f$ | 1000 | $\mathrm{kg\,m^{-3}}$ |
| Fluid thermal conductivity | $\lambda_f$ | 0.65 | $\mathrm{W\,m^{-1}\,K^{-1}}$ |
| Fluid heat capacity | $c_f$ | 4000 | $\mathrm{J\,kg^{-1}\,K^{-1}}$ |
| Fluid viscosity | $\mu_f$ | $1.0 \times 10^{-3}$ | $\mathrm{Pa\,s}$ |
| Fluid permeability | $k$ | $1.0 \times 10^{-10}$ | $\mathrm{m^2}$ |

## 4.1 Heat transport in a fracture

The first application deals with groundwater flow and (diffusive and advective) heat transport in a fracture, i.e. a TH application.
We assume a fracture which is fully saturated with water ($n = 1$), having homogeneous and isotropic properties. Groundwater
flow in the fracture is assumed to be unidirectional and the average velocity is considered constant throughout the length of
the flow field. The initial temperature is set to zero. At time $t = 0$, a sudden increase in temperature is imposed along the inlet
boundary of the medium ($T = T_0$). The problem can be formulated as a semi-infinite medium with a point source at the inlet
boundary ($T = T_0 H(x = 0, t)$ with $H(t)$ being the Heaviside function). Under these assumptions, Ogata and Banks (1961)
gave an analytical solution for the variation of the temperature as a function of the position and time as:

$$T(x,t) = \frac{T_0}{2} erfc\left(\frac{x - v_x t}{\sqrt{4kt}}\right) + \frac{T_0}{2} \exp\left(\frac{v_x x}{k}\right) erfc\left(\frac{x + v_x t}{\sqrt{4kt}}\right) \tag{41}$$

For the numerical simulation, we consider a fracture of length $L = 100\,\mathrm{m}$, which has been discretised into 1000 line elements
of equal length and subjected to an imposed pressure gradient of $\nabla p_f = 3\,\mathrm{Pa\,m^{-1}}$. All material properties are listed in Table 1.

Given the parameters considered, a constant fluid velocity ($v_x = 3.0 \times 10^{-7}\,\mathrm{m\,s^{-1}}$) is obtained. Initial conditions for the
pressure and temperature are $p_f = 0.1$ MPa and $T = 0$ °C respectively. In this benchmark, we compare the evolution of
the temperature field (normalised by the inlet temperature value) at four different points along the length of the fracture
with the analytical solution as given by Eq. 41. Figure 4 shows the comparison between model results (red curves) and the
corresponding analytical solutions (black circles) during the entire simulation time. There is a general good fit in terms of
the temporal evolution of the advected front at all location along the fracture plane, with the modelled thermal front moving
slightly faster as visible from the earlier stepping at the observation points. Based on the obtained results, we can conclude that
the TH numerical implementation is accurate for practical applications.

## 4.2 Flow in a fractured porous medium

The purpose of this application is to test the numerical implementation of the formulation in the presence of discrete fractures.
At this purpose, we refer to a relative common benchmark dealing with uniform, steady-state flow in a porous medium locally
disturbed by the presence of a fracture (Strack, 1982; Watanabe, 2011). The original problem formulation considers a semi-
infinite two-dimensional horizontal section with an embedded fracture located symmetrically at its centre. Uniform flow is





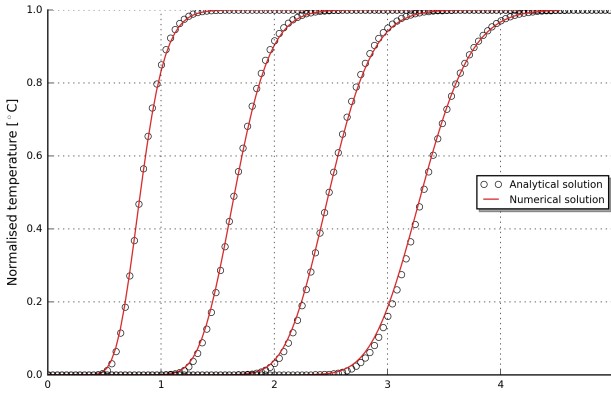

**Figure 4.** Comparison of numerical results (red curves) and analytical solutions (black circles) for heat transport in a fracture. The figure shows the temperature evolution at different positions along the fracture as a function of time.

**Table 2.** Material properties the example of flow in a fractured porous medium

|  | Property name | Symbol | Value | SI unit |
|---|---|---|---|---|
| Fracture | Angle | $\alpha$ | 45 | ° |
|  | Length | $L$ | 2 | m |
|  | Aperture | $h$ | 0.05 | m |
|  | Permeability | $k^f$ | $1.0 \times 10^{-10}$ | m$^2$ |
|  | Viscosity | $\mu_f^f$ | $1.0 \times 10^{-3}$ | Pa s |
| Porous medium | Porosity | $n$ | 0.15 | - |
|  | Permeability | $k$ | $1.0 \times 10^{-12}$ | m$^2$ |
|  | Viscosity | $\mu_f$ | $1.0 \times 10^{-3}$ | Pa s |

maintained by imposing a specific discharge ($q_0$) from the left boundary inside the domain, the value of which is kept constant and equal to $q_0 = 1.0 \times 10^{-4}$ m s$^{-1}$. The fracture is consider to extend infinitely along the direction normal to the plane, while being of finite along-plane length (L), with its middle point located exactly at the centre of the domain. It has a width which can be varied along its length (though it is assumed to remain constant in the following), an it is inclined with respect to the

5    model boundary by a constant angle ($\alpha$).

We have extended the original formulation to a three dimensional case, see Fig. 5 for the model geometry and boundary conditions. The setup of the problem comprises a three dimensional, quadrilateral box ($10 \times 10 \times 1$ m in x-y-z-directions respectively) representing the porous medium. The fracture is implemented as a two dimensional surface cutting entirely the model along the vertical and having a finite horizontal length of $L = 2$ m. It is inclined by an angle $\alpha = 45$° with respect to the

10    model boundaries. In order to maintain a constant discharge from the left to the right of the model, constant pressure boundary conditions are imposed along the same boundaries thus resulting in a constant pressure gradient along the x-axis. The value of the imposed pressure gradient ($\Delta p = 1$ MPa) has been enforced so to match the value of specific discharge of the original



problem as derived by (Strack, 1982), thus permitting a direct comparison between the analytical and numerical solution. No flow conditions are imposed along the other boundaries. We assume a laminar flow in the fracture plane, and pressure variations across its width are neglected. All relevant parameters are summarised in Table 2.

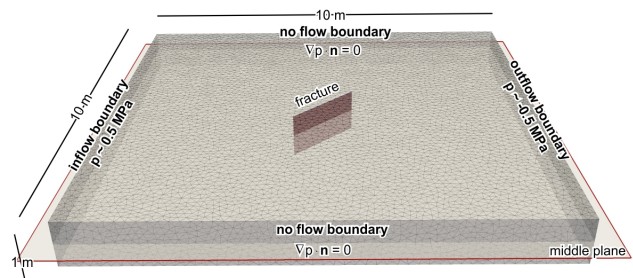

**Figure 5.** Geometry and boundary condition for the benchmark case of groundwater flow in a fractured porous medium.

Strack (1982) provided an analytical solution for the one dimensional version of the problem derived in terms of a complex
potential flow as:

$$\Omega = -A\sqrt{(Z-1)(Z+1)} + AZ - \frac{1}{2}q_0 L e^{i\alpha} Z \tag{42}$$

where $q_0$ is the magnitude of the flow occurring within the model domain (assumed uniform), $Z = \frac{z - \frac{1}{2}(z_1 + z_2)}{\frac{1}{2}(z_2 - z_1)}$ is the fracture-related dimensionless variable ($z_1$ and $z_2$ being the endpoints of the fracture), and $A = \frac{\frac{1}{2}K_f h}{K_m L + K_f h} q_0 L \cos\alpha$.

The numerical solution of the three dimensional problem has been obtained by solving a steady state flow problem within
the matrix-fracture domain. We plot the computed pressure distribution together with the outline of the geometry of the fracture in Fig. 6.

The presence of the discrete fracture disturbed the uniform horizontal flow in a close domain around its location. There, the isolines of constant pressure are distorted, and results in a faster flow preferentially oriented parallel to the fracture plane. In order to test the reliability of the numerical solution, we compare the computed pore pressure extracted along a diagonal cross
section cutting the model domain along its bottom left-top right corners with the pore pressure derived from the analytical solution. The comparison shows a perfect fit between the two results, thus proving the applicability of the discrete fracture approach as implemented in the current formulation.

The results obtained from a variation of the above described problem by considering two self-intersecting fractures embedded in a two-layered matrix are presented as Supplementary Information material.

**4.3   3D oedometer test**

In the following test, we consider a cube of porous medium with edges of $1$ m. The cube is subjected to axial loading with constant solid velocity $v_x$ under the conditions of an oedometer test (see setup in Fig. 7). The porous material undergoes

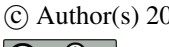



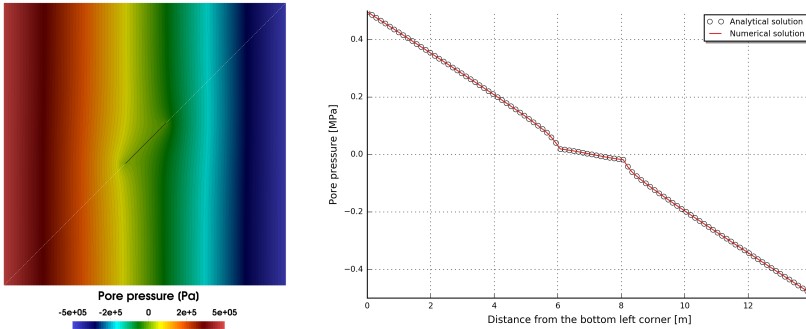

**Figure 6.** Isolines of pressure computed from the 3D numerical simulation extracted along a horizontal plane cutting the model domain. Comparison between simulated (continuous red curve) and analytical derived (empty black circles) pressure distribution along a line through the model.

**Table 3.** Mechanical properties for the oedometer benchmark.

| Property name | Symbol | Value | Unit |
|---|---|---|---|
| Bulk modulus | $K$ | $2.0 \times 10^3$ | MPa |
| Shear modulus | $G$ | $2.0 \times 10^3$ | MPa |
| Cohesion | $C$ | 1 | MPa |
| Friction angle | $\varphi$ | 20 | ° |
| Dilation angle | $\psi$ | 0, 10 or 20 | ° |
| Velocity | $v_x$ | $1.0 \times 10^{-5}$ | $\mathrm{m\,s^{-1}}$ |
| Edge of the cube | $L$ | 1 | m |

continuous loading, and it behaves elastically until the strength of the material is reached. From this time on, the material undergoes plastic loading. The elastoplastic constitutive laws adopted for this simple problem formulation is the Drucker-Prager plasticity model. The Drucker-Prager yield envelope is a smoother version of the classical Mohr-Coulomb failure criterion. Under these conditions, an analytical solution for the stress state of the porous material can be derived as described in

5   Appendix A and serves here as verification of the numerical implementation of the elastoplastic constitutive laws. The physical properties used for this benchmark are summarised in Table 3.

Figure 7 shows the evolution of stress for an associative (red dots) and two non-associative (dilation angle different from friction angle) plastic potentials (blue and green dots). The results exhibit a perfect agreement between the analytical solution and the numerical prediction. Based on these observations, we can conclude on the validity of the implementations for the

10   elasto-plastic constitutive laws and for the return-map algorithm as implemented in the current formulation.

### 4.4 Prototype of multi-fractured geothermal reservoir

In this example, we present a set up inspired by a typical geothermal reservoir application. The model aims at simulating the thermal and hydraulic configuration of the reservoir during operational activities (injection and production) spanning a life time





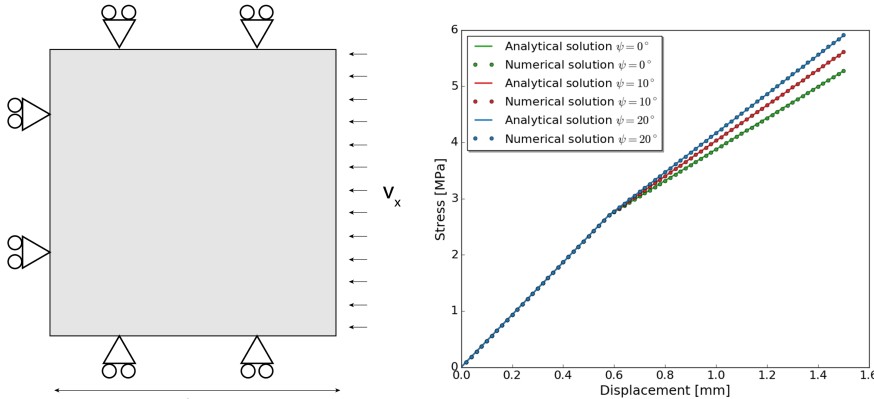

**Figure 7.** Problem formulation and results of the oedometer benchmark. Panel (a) shows the problem formulation. Panel (b) illustrates the results for different dilation angles with the stress-displacement curves.

of the reservoir of approximately 100 years. The model consists of four different geological formations, two units representing the target reservoir plus an upper and lower formation acting as cap rocks. The extent of the model domain is $10 \times 10 \times 3 \, \text{km}$ in the x-y-z- directions. The target reservoir is located at a depth of approximately $4.6 \, \text{km}$ below sea level, and is cut by a natural fault showing a slip of some hundreds meters at depths of relevance in the reservoir. A doublet system is integrated

in the model, consisting of an injection and a production well. The open hole section of the two wells is kept parallel and extend for approximately 1,5 km horizontally in the reservoir. The open hole section of the wells has been integrated as one dimensional finite elements and homogenisation of the resulting governing equations is done by considering the surface area of the well bore as scaling parameter. A system of hydraulically stimulated fractures is also considered to enhance the hydraulic connection between the two wells along their horizontal sections. The multi-frac system is intended to represent an hydraulic

stimulation campaign prior to reservoir exploitation. There are a total of ten fractures, equally spaced every $100 \, \text{m}$ and all sharing the same geometry and material properties. A schematic representation of the main geometry is illustrated in Fig. 8, and all properties are listed in Table 4.

Operational activity is simulated by injecting water at a fixed temperature, $T_{in} = 55 \, °\text{C}$, and at a constant rate $q_{in} = 30 \, \text{L s}^{-1}$ for the entire simulation period. Production rates are kept equal to injection rates through the operation. Figure 8 illustrates

the reservoir state at the end of the simulation (an animation of the full evolution of the system is provided as Supplementary Information material). Figure 8 nicely illustrates the evolution of the reservoir temperature as resulting from the dynamics of interactions between the different components of the system (i.e. reservoir matrix, fractures and geothermal wells) and shows how the simulator can be effectively applied to 3D modelling of complex reservoirs.



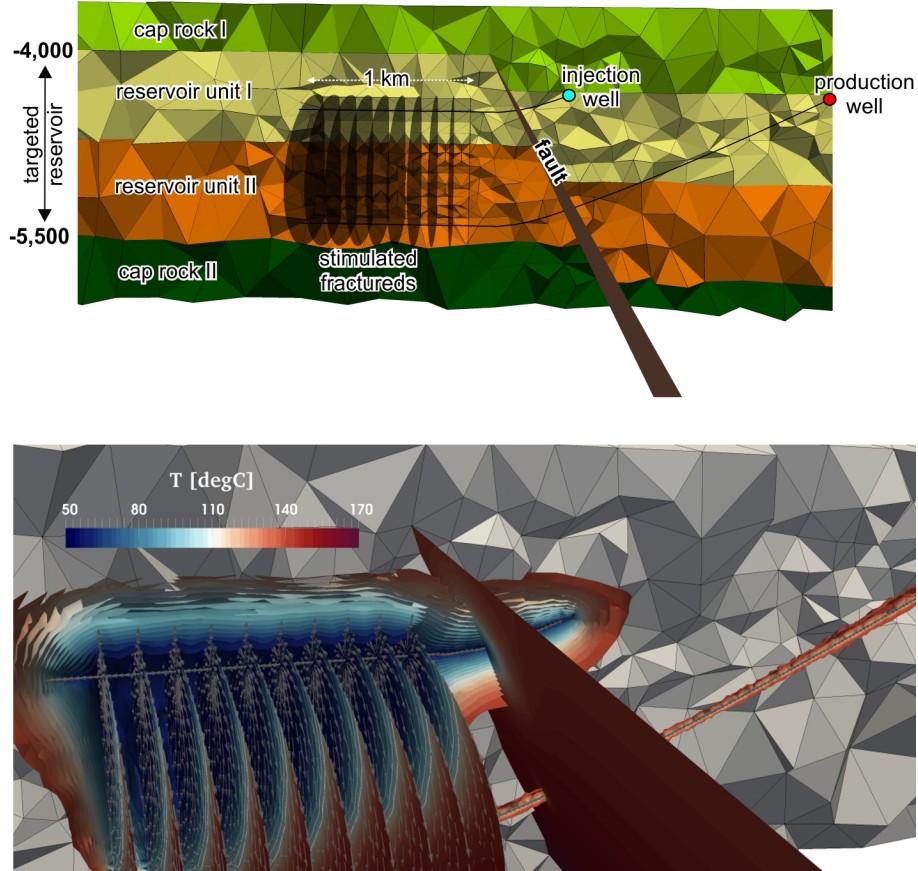

**Figure 8.** Problem formulation and results of the prototype of multi-fractured geothermal reservoir. Panel (a) shows the geometry and setup of the simulation and panel (b) the distribution of the temperature after approximately 100 years of production.

## 4.5 Thermo-poroelastic response of a synthetic geothermal doublet

This example considers operations of a synthetic geothermal doublet within a low permeability geological formation with induced fractures located at the two operational wells. The model aims at describing the thermo-poroelastic response of the reservoir due to geothermal operations. The extent of the model domain is $500 \times 500 \times 200$ m in the x-y-z- directions. The depth of the target reservoir is approximatively $4$ km below sea level. Pore pressure and temperature distributions are assumed to be homogeneous at the beginning of the simulation, and equal to $40$ MPa and $150\,°C$ respectively. A regional stress field is applied as background stress, to simulate a normal faulting regime with the following magnitudes:





**Table 4.** Material properties for the multi-frac reservoir application.

| Unit | Property name | Symbol | Value | SI unit |
|---|---|---|---|---|
| | Porosity | $n$ | 0.01 | - |
| | | $k_x$ | $1.0 \times 10^{-20}$ | $\text{m}^2$ |
| | Permeability | $k_y$ | $1.0 \times 10^{-20}$ | $\text{m}^2$ |
| Cap rock I | | $k_z$ | $0.25 \times 10^{-20}$ | $\text{m}^2$ |
| | Fluid modulus | $K_f$ | $1.0 \times 10^8$ | Pa |
| | Fluid viscosity | $\mu_f$ | $3.0 \times 10^{-4}$ | Pa s |
| | Rock density | $\rho_s$ | 2650 | $\text{kg m}^{-3}$ |
| | Rock thermal conductivity | $\lambda_s$ | 4 | $\text{W m}^{-1}\text{K}^{-1}$ |
| | Rock heat capacity | $c_s$ | 920 | $\text{J kg}^{-1}\text{K}^{-1}$ |
| | Porosity | $n$ | 0.01 | - |
| | | $k_x$ | $1.0 \times 10^{-20}$ | $\text{m}^2$ |
| | Permeability | $k_y$ | $1.0 \times 10^{-20}$ | $\text{m}^2$ |
| Cap rock II | | $k_z$ | $1.0 \times 10^{-20}$ | $\text{m}^2$ |
| | Fluid modulus | $K_f$ | $1.0 \times 10^8$ | Pa |
| | Fluid viscosity | $\mu_f$ | $3.0 \times 10^{-4}$ | Pa s |
| | Rock density | $\rho_s$ | 2650 | $\text{kg m}^{-3}$ |
| | Rock thermal conductivity | $\lambda_s$ | 2.31 | $\text{W m}^{-1}\text{K}^{-1}$ |
| | Rock heat capacity | $c_s$ | 1380 | $\text{J kg}^{-1}\text{K}^{-1}$ |
| | Porosity | $n$ | 0.15 | - |
| | | $k_x$ | $1.28 \times 10^{-15}$ | $\text{m}^2$ |
| | Permeability | $k_y$ | $1.28 \times 10^{-15}$ | $\text{m}^2$ |
| Reservoir unit I | | $k_z$ | $3.2 \times 10^{-16}$ | $\text{m}^2$ |
| | Fluid modulus | $K_f$ | $1.5 \times 10^9$ | Pa |
| | Fluid viscosity | $\mu_f$ | $3.0 \times 10^{-4}$ | Pa s |
| | Rock density | $\rho_s$ | 2650 | $\text{kg m}^{-3}$ |
| | Rock thermal conductivity | $\lambda_s$ | 3.18 | $\text{W m}^{-1}\text{K}^{-1}$ |
| | Rock heat capacity | $c_s$ | 920 | $\text{J kg}^{-1}\text{K}^{-1}$ |
| | Porosity | $n$ | 0.005 | - |
| | | $k_x$ | $9.87 \times 10^{-17}$ | $\text{m}^2$ |
| | Permeability | $k_y$ | $9.87 \times 10^{-17}$ | $\text{m}^2$ |
| Reservoir unit II | | $k_z$ | $2.4675 \times 10^{-17}$ | $\text{m}^2$ |
| | Fluid modulus | $K_f$ | $1.0 \times 10^8$ | Pa |
| | Fluid viscosity | $\mu_f$ | $3.0 \times 10^{-4}$ | Pa s |
| | Rock density | $\rho_s$ | 2650 | $\text{kg m}^{-3}$ |
| | Rock thermal conductivity | $\lambda_s$ | 2.31 | $\text{W m}^{-1}\text{K}^{-1}$ |
| | Rock heat capacity | $c_s$ | 1380 | $\text{J kg}^{-1}\text{K}^{-1}$ |
| | Porosity | $n$ | 1 | - |
| | Aperture | $h$ | $1.0 \times 10^{-2}$ | m |
| | | $k_x$ | $1.0 \times 10^{-15}$ | $\text{m}^2$ |
| Fault | Permeability | $k_y$ | $1.0 \times 10^{-15}$ | $\text{m}^2$ |
| | | $k_z$ | $1.0 \times 10^{-15}$ | $\text{m}^2$ |
| | Fluid modulus | $K_f$ | $2.5 \times 10^9$ | Pa |
| | Fluid viscosity | $\mu_f$ | $3.0 \times 10^{-4}$ | Pa s |
| | Fluid density | $\rho_s$ | 1148 | $\text{kg m}^{-3}$ |
| | Fluid thermal conductivity | $\lambda_s$ | 0.65 | $\text{W m}^{-1}\text{K}^{-1}$ |
| | Fluid heat capacity | $c_s$ | 4193.5 | $\text{J kg}^{-1}\text{K}^{-1}$ |
| | Porosity | $n$ | 1 | - |
| | Aperture | $h$ | $2.28 \times 10^{-4}$ | m |
| | | $k_x$ | $4.33 \times 10^{-9}$ | $\text{m}^2$ |
| Fractures | Permeability | $k_y$ | $4.33 \times 10^{-9}$ | $\text{m}^2$ |
| | | $k_z$ | $4.33 \times 10^{-9}$ | $\text{m}^2$ |
| | Fluid modulus | $K_f$ | $2.5 \times 10^9$ | Pa |
| | Fluid viscosity | $\mu_f$ | $3.0 \times 10^{-4}$ | Pa s |
| | Fluid density | $\rho_s$ | 1148 | $\text{kg m}^{-3}$ |
| | Fluid thermal conductivity | $\lambda_s$ | 0.65 | $\text{W m}^{-1}\text{K}^{-1}$ |
| | Fluid heat capacity | $c_s$ | 4193.5 | $\text{J kg}^{-1}\text{K}^{-1}$ |



**Table 5.** Material properties for the thermo-poroelastic response of a geothermal doublet

| Unit | Property name | Symbol | Value | SI unit |
|---|---|---|---|---|
| | Porosity | $n$ | 0.1 | - |
| | Permeability | $k$ | $1.0 \times 10^{-15}$ | $\mathrm{m}^2$ |
| Reservoir | Fluid modulus | $K_f$ | $1.0 \times 10^{8}$ | Pa |
| rock | Rock density | $\rho_s$ | 2600 | $\mathrm{kg\,m^{-3}}$ |
| | Rock thermal conductivity | $\lambda_s$ | 3 | $\mathrm{W\,m^{-1}\,K^{-1}}$ |
| | Rock heat capacity | $c_s$ | 950 | $\mathrm{J\,kg^{-1}\,K^{-1}}$ |
| | Porosity | $n$ | 1 | - |
| | Aperture | $h$ | $1.0 \times 10^{-2}$ | m |
| | Permeability | $k$ | $8.333 \times 10^{-10}$ | $\mathrm{m}^2$ |
| Fractures | Fluid modulus | $K_f$ | $2.5 \times 10^{9}$ | Pa |
| | Fluid density | $\rho_f$ | 1000 | $\mathrm{kg\,m^{-3}}$ |
| | Fluid thermal conductivity | $\lambda_f$ | 0.65 | $\mathrm{W\,m^{-1}\,K^{-1}}$ |
| | Fluid heat capacity | $c_f$ | 4200 | $\mathrm{J\,kg^{-1}\,K^{-1}}$ |

– Vertical stress $S_1 = 100$ MPa in the z-direction.

– Maximum horizontal stress $S_2 = 90$ MPa in the y-direction.

– Minimum horizontal stress $S_3 = 50$ MPa in the x-direction.

Two hydraulic fractures are considered for this doublet system and represent the impacts of a prior hydraulic stimulation campaign to enhance the productivity of the reservoir. Given the in situ stress conditions, these two hydraulic fractures are orthogonal to the minimum horizontal stress. They are implemented as squares with edges of $100$ m. The distance between the fractures is $200$ m.

In this simulation, we consider additional non linear effects as related to imposed variations in the evolution of the fluid (i.e. fluid density and viscosity) and rock properties (i.e. porosity and permeability) as a function of the evolution in the state of the reservoir reservoir during operational activities. Changes in porosity are controlled by Eq. 22, where we neglected second-order terms for this specific application, while the evolution in the rock permeability is governed by a classical Kozeny-Carman like relation as:

$$k = A \frac{n^3}{(1-n)^2} \tag{43}$$

where the coefficient $A$ includes information about the pores and grain geometries and can be expressed via the initial value of porosity and permeability: $A = k_0 \frac{(1-n_0)^2}{n_0^3}$.

Figure 9 illustrates the model geometry.

Geothermal operations consist of injecting water at $T_{in} = 70\ {}^\circ$C and at a constant rate $q_{in} = 5\ \mathrm{L\,s^{-1}}$ and producing geothermal fluid at the same rate. Pore pressure and temperature are kept constant on all sides of the model as boundary conditions.

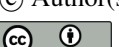


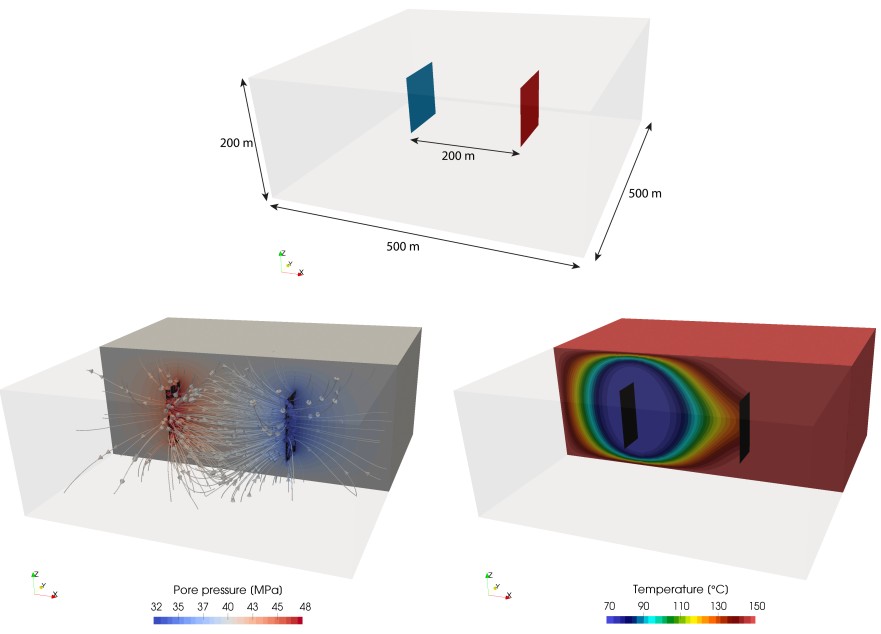

**Figure 9.** Problem formulation and results of the thermo-poroelastic response of a geothermal doublet. Panel (a) shows the problem formulation. Panels (b) and (c) illustrates the pore pressure (with fluid velocity) and temperature distributions respectively after 50 years of operations.

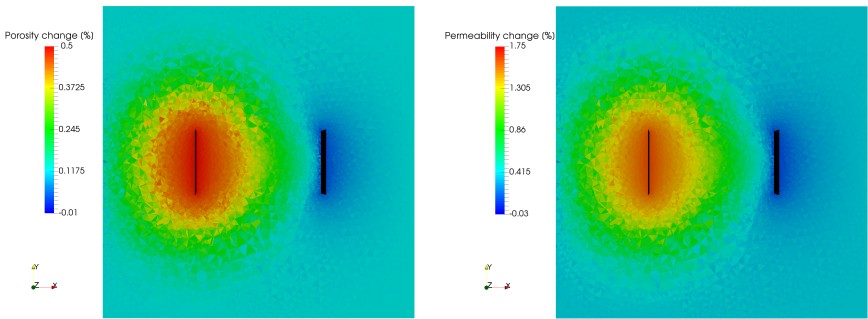

**Figure 10.** Horizontal slices of the model illustrating the changes in transport properties. Panel (a) shows the distribution of the changes in porosity and panel (b) the changes in permeability after 50 years of operations.

All faces are considered as sliding faces for boundary conditions in terms of displacements. The simulation covers a complete time of 50 years of geothermal operations. All physical properties used for this example are summarised in Table 5.

Figure 9 shows the distribution of pore pressure and temperature at the end of the simulation. It can be observed that colder temperature than the reservoir temperature are produced at the right well after 50 years of operations.




Figure 10 shows horizontal slices at the centre of the domain illustrating the changes in porosity and permeability due to the applied changes in strain, pore pressure and temperature.

This example illustrates the simulator capability of solving a full Thermo-Hydro-Mechanical coupled problem as relevant for geothermal reservoir applications. At the same time, the formulation adopted to simulate the evolution in time and space

of the system properties, enables to quantify the impact of the mechanical alteration induced by geothermal operations (injection/production of fluid and applied injection temperature) on both fluid and heat flows within the reservoir via transport.

## 5   Conclusions

In this paper, we presented a novel but robust simulator for modelling coupled THM processes within fractured rocks, with specific focus on reservoir applications. The code, GOLEM, relies on an open source massive parallel Finite Element based

numerical framework (MOOSE) to solve for the coupled problem. It makes use of a fully implicit approach to treat the nonlinear coupling among the different processes and their feedback effects on fluid and rock properties, thus providing higher numerical stability in the context of nonlinear problems. Geological heterogeneities, i.e. discrete fractures and fault zones, are taken into account in our formulation. The latter are represented as finite element of lower geometrical dimension, which allows to model focused fluid and heat flows on fractures and faults planes or well paths. The capability and robustness of the

simulator has been illustrated by means of five numerical examples by increasing progressively the coupling and geometrical complexity of the considered problem formulations.

Improving the reliability of predictions made for geothermal operations at the field scale requires a better description of the physical phenomena which can alter the reservoir productivity as well as the sustainability of the geothermal operations. In this respect, the current framework provides a powerful tool to analyse the dynamic behaviour of fractured reservoirs during

geothermal operations.

Ongoing activities are towards integration of the details of the mechanical description of the geological discontinuities (faults and fractures) either by means of a discrete (XFEM approach) or by a continuous (such as phase field) approach. Such a feature will help to better describe the dynamics deformation in heterogeneous rocks, including localization and evolution along fault zones, and will also permit to quantitative integrate feedbacks on the hydraulic and thermal behaviours of such geological

structures. The description of such processes would help at forecasting environmental impacts of reservoir operations such as induced seismicity of dynamic reactivation of faults during operational activities.

## 6   Code and data availability

The source code as well as the input files of the five numerical examples presented in this paper, plus a suite of specific benchmark cases, are available upon request by contacting one of the two authors.





## Appendix A:  Analytical solution for the oedometer benchmark

During elastic loading with constant velocity $v_x$, the strain evolves as:

$$\epsilon_{xx} = \frac{v_x t}{L}, \quad \epsilon_{yy} = \epsilon_{zz} = 0 \tag{A1}$$

because of the no displacement boundary conditions in the $y$- and $z$-directions. The stress during elastic loading therefore

reads:

$$\sigma_{xx} = (\lambda + 2G)\frac{v_x t}{L}, \quad \sigma_{yy} = \sigma_{zz} = \lambda \frac{v_x t}{L}. \tag{A2}$$

With these expressions of stress, the invariants of the stress tensor, $J_1$ and $J_2$ can be written as:

$$J_1 = (3\lambda + 2G)\frac{v_x t}{L}, \quad J_2 = \frac{4}{3}\left(G\frac{v_x t}{L}\right)^2. \tag{A3}$$

The onset of plastic strain accumulation is reached when the yield function reaches 0 at a time noted $t_y$, that is $\mathcal{F}(t_y) = 0$.

By using the expressions of the stress invariants in Eq. A3, the time for onset of yielding can be expressed as:

$$t_y = \frac{LC\cos\varphi}{|v_x|\left(\frac{2}{\sqrt{3}}G - \sin(\varphi)(\lambda + \frac{2}{3}G)\right)}. \tag{A4}$$

The increment of plastic strain accumulation can be expressed by reinjecting the expressions of the stress invariant and Eq. 40 into Eq. 20:

$$\begin{aligned}
\Delta\epsilon_{xx}^{*p} &= \frac{\Delta\gamma}{\sqrt{3}}\left(-1 + \frac{\sin(\psi)}{\sqrt{3}}\right) \\
\quad \Delta\epsilon_{yy}^{*p} &= \frac{\Delta\gamma}{\sqrt{3}}\left(\frac{1}{2} + \frac{\sin(\psi)}{\sqrt{3}}\right) \\
\Delta\epsilon_{zz}^{*p} &= \frac{\Delta\gamma}{\sqrt{3}}\left(\frac{1}{2} + \frac{\sin(\psi)}{\sqrt{3}}\right)
\end{aligned} \tag{A5}$$

The increment of stress during plastic accumulation can therefore be written:

$$\begin{aligned}
\Delta\sigma_{xx} &= (\lambda + 2G)\frac{v_x\Delta t}{L} + \Delta\gamma\left[\frac{2}{\sqrt{3}}\left(1 - \frac{\sin(\psi)}{\sqrt{3}}\right)G - \lambda\sin(\psi)\right] \\
\Delta\sigma_{yy} &= \lambda\frac{v_x\Delta t}{L} - \Delta\gamma\left[\frac{2}{\sqrt{3}}\left(\frac{1}{2} + \frac{\sin(\psi)}{\sqrt{3}}\right)G + \lambda\sin(\psi)\right] \\
\quad \Delta\sigma_{zz} &= \lambda\frac{v_x\Delta t}{L} - \Delta\gamma\left[\frac{2}{\sqrt{3}}\left(\frac{1}{2} + \frac{\sin(\psi)}{\sqrt{3}}\right)G + \lambda\sin(\psi)\right]
\end{aligned} \tag{A6}$$

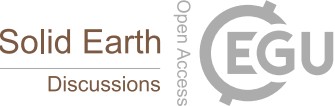



Furthermore, the Kuhn-Tucker conditions imply that the derivative over time of the yield function remains $0$ during plastic accumulation, that is, in incremental form:

$$\Delta \mathcal{F} = 0 \Leftrightarrow \frac{-1}{\sqrt{3}} \left( \Delta \sigma_{xx} - \Delta \sigma_{yy} \right)$$
$$+ \frac{\sin(\varphi)}{3} \left( \Delta \sigma_{xx} + 2\Delta \sigma_{yy} \right) = 0. \tag{A7}$$

By injecting Eq. A6 into Eq. A7, one finally obtains the value for the increment of plastic multiplier as:

$$\Delta \gamma = \frac{\frac{2}{\sqrt{3}} G - \sin(\varphi) \left( \lambda + \frac{2}{3} G \right)}{G + \frac{2}{3} \left( \lambda + G \right) \sin(\varphi) \sin(\psi)} \frac{|v_x| \Delta t}{L}. \tag{A8}$$

The solution for plastic strain and stress can therefore be integrated as:

$$\epsilon_{xx}^{*p}(t) = \frac{\gamma(t)}{\sqrt{3}} \left( -1 + \frac{\sin(\psi)}{\sqrt{3}} \right)$$
$$\epsilon_{yy}^{*p}(t) = \frac{\gamma(t)}{\sqrt{3}} \left( \frac{1}{2} + \frac{\sin(\psi)}{\sqrt{3}} \right) \tag{A9}$$
$$\epsilon_{zz}^{*p}(t) = \frac{\gamma(t)}{\sqrt{3}} \left( \frac{1}{2} + \frac{\sin(\psi)}{\sqrt{3}} \right)$$

$$\sigma_{xx}(t) = (\lambda + 2G)\frac{v_x t}{L} + \gamma(t)\left[ \frac{2}{\sqrt{3}} \left( 1 - \frac{\sin(\psi)}{\sqrt{3}} \right) G - \lambda \sin(\psi) \right]$$
$$\sigma_{yy}(t) = \lambda \frac{v_x t}{L} - \gamma(t)\left[ \frac{2}{\sqrt{3}} \left( \frac{1}{2} + \frac{\sin(\psi)}{\sqrt{3}} \right) G + \lambda \sin(\psi) \right] \tag{A10}$$
$$\sigma_{zz}(t) = \lambda \frac{v_x t}{L} - \gamma(t)\left[ \frac{2}{\sqrt{3}} \left( \frac{1}{2} + \frac{\sin(\psi)}{\sqrt{3}} \right) G + \lambda \sin(\psi) \right]$$

with the plastic multiplier as:

$$\gamma(t) = \frac{\frac{2}{\sqrt{3}} G - \sin(\varphi) \left( \lambda + \frac{2}{3} G \right)}{G + \frac{2}{3} \left( \lambda + G \right) \sin(\varphi) \sin(\psi)} \frac{|v_x| t}{L}. \tag{A11}$$

15 *Author contributions.* M. Cacace and A.B. Jacquey developed the model code, designed the examples presented in the manuscript and performed the simulations. Both authors also share the preparation and writing of the manuscript.

*Competing interests.* The authors declare that they have no conflict of interest.





*Acknowledgements.* The authors would like to thank the developers team of the MOOSE framework as well as the members of the MOOSE users community for their helpful support and guidance for code development using the MOOSE framework. This work was funded by the Helmholtz Association as part of the Helmholtz Portfolio Geo-Energy project.



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
