# Peer review of "Flexible parallel implicit modelling of coupled Thermal-Hydraulic-Mechanical processes in fractured rocks"

_Solid Earth, 2017_

## Referee Comment (RC1) · M. Veveakis (Referee) · 13 Jun 2017

My sincere apologies for the extremely late comment. I have read the manuscript with great interest and I think it does fit the scope of the journal and should be published following some minor to moderate revisions. My main comment concerns the presentation of the mechanical model, with the detailed emphasis given in the algorithmic implementation of the plasticity algorithm without being used at all in this manuscript. This confuses the reader who expects to see an application using the return map described in detail without delivering it.

[Figure]

i would recommend the authors to remove the discussion about plasticity, keep the model elastic and put more emphasis in the excellent geothermal example of paragraph 4.5. In its present form, the paper is weak in the sense that most of the benchmarks described already exist in the tensor mechanics module of MOOSE and as such they do not really deserve to be in the main text of a novel contribution. The geothermal example however, is novel and impressive and deserves its own section and some more detail to be added.

On a minor note, it would be nice to acknowledge other modules that are doing fully coupled THMC with thermo-hydro-chemically sensitive plasticity, like REDBACK. We have just released the first works that I think deserve to be mentioned in the literature review section of this contribution:

1. Poulet T. and E. Veveakis, 2016. A viscoplastic approach for pore collapse in saturated soft rocks using REDBACK: an open-source parallel simulator for Rock mEchanics with Dissipative feedBACKs, Computers and Geotechnics, 74, 211-221, doi:10.1016/j.compgeo.2015.12.015

2. Poulet T., M. Paesold and E. Veveakis, 2017. Multiphysics modelling for fault mechanics using REDBACK: A parallel open–source simulator for tightly coupled problems, Rock Mech. Rock Eng., 50(3), 733-749, doi: 10.1007/s00603-016-0927-y

I do not expect the authors to need more than a few weeks to address these points. Other than that the manuscript is very well written and should get published in Solid Earth.

Regards, Manolis Veveakis
* * *

---

## Referee Comment (RC2) · J. Taron (Referee) · 19 Jun 2017

I must apologize for the delay in this review. I thoroughly enjoyed reading the article. Presentation is clear and well-written. In the final analysis, your story is worthwhile and deserving of publication. I have some revisions.

minor:

1) Change THOUGH to TOUGH on page 2, line 20.

[Figure]

2) Strain decomposition is introduced before defined. Suggest moving paragraph at page 8, line 5 before equation 12.

3) Page 9, line 5. You state that the biot/bulk expansion term is of second order. Please explain or provide reference.

4) Your presentation of the energy balance is somewhat topical relative to the other formulations in the paper. You should either begin from internal energy, or cite the origin of equation 23 and note the assumptions required to attain it. Also, the dissipation term added ad-hoc to equation 25 should instead be present and justified in 23.

5) Figure 7: Colors are fine, but different symbols or inlayed arrows should be added. Figure 8: Should probably specify that you ignore contours beyond 170.

6) Please specify boundary conditions in Section 4.5 or in figure 8. Also, in the simulation with true fluid properties, you must have equilibrated the system with those properties, otherwise you would see some non-linear porosity-change depth variation globally. I presume you computed and restarted, but please state this.

7) Also section 4.5. How are the wells treated? As lower-dimensional "fractures"?.

general:

1) It is unclear to me which portions of your work are extensions to MOOSE, and which are already present. This is important and needs to be stated clearly.

2) In most reservoirs I have worked with, permeability change is dominated by the behavior of fractures, not by porous mechanics (i.e. Kozeny-Carman), which are minimally important. Perhaps this reservoir is not highly fractured, or perhaps you only interesting in displaying porous behavior for the sake of illustration. In either case, you need to clarify the intent of your assumptions and their shortcomings.

3) My last and largest comment is in agreement with the previous review. Essentially, you have provided details of capabilities that are then never utilized in the final analysis.

I understand this temptation, but it detracts from the story you are trying to tell. The only useage of plasticity in the paper is to demonstrate an oedometer problem. Either add greater complexity to your final problem (I understand there is a fair amount required), or remove plasticity from the discussion. Alternatively, you might choose a validation more along the lines of a Mandel-Cryer type problem (preferably at interesting temperature and pressure and with properties computed internally), which is a fundamental aspect of your final elastic simulation and rounds out the other validations.

---

## Author Comment (AC1) · 11 Jul 2017

General comment: My sincere apologies for the extremely late comment. I have read the manuscript with great interest and I think it does fit the scope of the journal and should be published following some minor to moderate revisions.

Answer to the general comment: We would like to thank the reviewer for his comments. We have systematically addressed all of his points, as explained below. When we did believe the comments from Reviewer #1 required changes

in the manuscript, we have followed his suggestions and implemented those changes accordingly (these are all marked in the revised version, see the file ca-cace_jacquey_solid_earth_tracked_changes.pdf). We have provided detailed answers for those comments that either we did not agree with the reviewer's point of view or we did not consider requiring any change to the manuscript.

Point 1: My main comment concerns the presentation of the mechanical model, with the detailed emphasis given in the algorithmic implementation of the plasticity algorithm without being used at all in this manuscript. This confuses the reader who expects to see an application using the return map described in detail without delivering it. I would recommend the authors to remove the discussion about plasticity, keep the model elastic and put more emphasis in the excellent geothermal example of paragraph 4.5. In its present form, the paper is weak in the sense that most of the benchmarks described already exist in the tensor mechanics module of MOOSE and as such they do not really deserve to be in the main text of a novel contribution. The geothermal example however, is novel and impressive and deserves its own section and some more detail to be added.

Answer to Point 1: We acknowledge the point raised by the reviewer, though we would like to add a short discussion on some aspects on Reviewer#1. First of all, concerning his main comment on the abswence in the manuscript on a practical example of the return map procedure we should add that we did present an application dealing with the plasticity (and the return mapping algorithm we implemented in our simulator) and we validated our implementation against an analytical solution derived in details in the Appendix. We nevertheless acknowledge the point from Reviewer#1 in the sense that in its present form, the organization of the results in the current manuscript might have read a bit confusing and we have re-organized the results part according to his suggestions, and by also taking into consideration the suggestions from Reviewer#2. Now we have provided a clear distinction between benchmark test cases which we make use to compare the reliability and reproducibility of the simulator against existing

analytical solutions in addressing specific coupling (as commonly done in the modelling community) and reservoir applications. For each case we have highlighted the processes targeted to ease the understanding of each example. All changes can be found in the revised manuscript accordingly. With respect to the second aspect of Reviewer#1 comment, we would like to add that we could not fully agree with Reviewer#1 in saying that the test cases presented are already part of the benchmark suite of the MOOSE framework. Indeed, while they can be considered as commonly adopted test cases, they have not been taken from the framework itself. This is why we did spend some time in the appendix to derive the analytical solution for the oedometer benchmark while using a Drucker-Prager plastic model. Specifically with respect to the applications focusing on the elastic and inelastic processes, we once again followed Reviewer#1 suggestions, and also take into account the comments from Reviewer#2, and we add an additional study case which increases the level of complexity of the reference oedometer test by additionally considering the hydromechanical response under undrained condition with poroelastoplastic coupling in the revised version. Following the reviewer's suggestions we have decided to dedicate a separate paragraph to the applied (reservoir) applications, both the hydrothermal multi-fractured and the thermo-hydro-mechanical applications. As a closing comment to our answer so far, we would like to spend some more words on the reason why we did consider important to maintain the description of the elastoplastic constitutive laws and their integration in the revised manuscript. The scope of the manuscript is to describe the full capability of the simulator we have developed. As stated in the introduction, attention was mainly focused on capturing the details of the non-linear feedbacks between thermal, hydraulic (pore pressure) and mechanical processes as of relevance in the context of reservoir applications in the presence of discrete fractures and/or major fault zones. In this regard, the latter component entail to systematically address and integrate the effect related to the (poro and thermo) elastic behaviour of the porous material, but also the irreversible one, e.g. inelastic processes. To implement inelastic behaviour in the context of porous media application requires to develop specific algorithmic features (in

our case the return map algorithm for non-associated plasticity) that are generally non trivial. This is the main reason we spent some efforts in, briefly explained how such algorithms have been integrated in our current numerical framework, in a way that would be reproducible by other scientists who might follow the same route. In this regard, the reviewer might be right in saying that the discussion provided might read long, but this has likely to do with him being familiar with the topic (in the context of numerical implementation). Indeed, we have already been contacted by other colleagues who were specifically interested to learn more details (than those provided in the manuscript) on our implementation of the inelastic routines. For this reason, we consider important to maintain the description of the algorithm in the manuscript (leaving it off would also hindered the reader and potential user of the numerical framework to fully appreciate the capability of Golem).

Point 2: On a minor note, it would be nice to acknowledge other modules that are doing fully coupled THMC with thermo-hydro-chemically sensitive plasticity, like RED-BACK. We have just released the first works that I think deserve to be mentioned in the literature review section of this contribution.

Answer to Point 2: We acknowledge the comment from Reviewer#1, and would like to apologise for our lack of referencing other works and had the expected reference in the revised version of the paper (see introduction in the revised version).

---

## Author Comment (AC2) · 11 Jul 2017

General comment: I must apologize for the delay in this review. I thoroughly enjoyed reading the article. Presentation is clear and well-written. In the final analysis, your story is worthwhile and deserving of publication. I have some revisions.

Answer to the general comment: We would like to thank the reviewer for his comments. We have addressed all of his point as detailed below. As done with the comments from Reviewer#1, all changes made in the revised text have been highlighted, and we

provide detailed answers to comments that we think did not require any change to the manuscript.

Point 1: Change THOUGH to TOUGH on page 2, line 20.

Answer to Point 1: Done.

Point 2: Strain decomposition is introduced before defined. Suggest moving paragraph at page 8, line 5 before equation 12.

Answer to Point 2: We are missing the point from the reviewer. In equation 12 we described the stress-strain relationship following Biot elasticity theory, but we make no reference to any inelastic component to the deformation. Only later these additional components are introduced, following an additive strain decomposition (equation 17), and we only make use of it in the following equations to integrate the inelastic process.

Point 3: Page 9, line 5. You state that the biot/bulk expansion term is of second order. Please explain or provide reference.

Answer to Point 3: We have some problems in understanding the comment of the reviewer. Indeed, in the sentence pointed out by the reviewer, we consider the last term of the RHS of equation 22 (those that are driven by the solid deformation velocity) as secondary terms (this comes from linear analysis of the infinitesimal orders of the different terms, see Biot (1959) for example). To this point, and in order to avoid any misunderstanding, following his suggestion we have added the latter reference to the revised version. Concerning the bulk thermal expansion terms, they are neither considered as secondary nor neglected, as they do appear in the second term in the RHS of the same equation 22.

Point 4: Your presentation of the energy balance is somewhat topical relative to the other formulations in the paper. You should either begin from internal energy, or cite the origin of equation 23 and note the assumptions required to attain it. Also, the dissipation term added ad-hoc to equation 25 should instead be present and justified

in 23.

Answer to Point 4: We have added the reference to the revised version of the paper.

Point 5: Figure 7: Colors are fine, but different symbols or inlayed arrows should be added. Figure 8: Should probably specify that you ignore contours beyond 170.

Answer to Point 5: We have added different inlays symbols to Figure 7. Regarding Figure 8, 170 was the maximum temperature considered in the reservoir. We apologize for the missing information and we have added it to the revised version of the manuscript.

Point 6: Please specify boundary conditions in Section 4.5 or in figure 8. Also, in the simulation with true fluid properties, you must have equilibrated the system with those properties, otherwise you would see some non-linear porosity-change depth variation globally. I presume you computed and restarted, but please state this.

Answer to Point 6: We have added all boundary conditions for the simulation of the multifrac reservoir. The reviewer is right while saying that the initial condition for the transient response was based on a steady simulation (in order to equilibrate the fluid material properties at the start of the simulation). We have inserted this missing information in the revised text. However, this was not the case for the thermo-poroelastic case, where pore pressure and temperature were considered initially constant (as stated in the text) and we make use of a background stress state to initialize the model.

Point 7: Also section 4.5. How are the wells treated? As lower-dimensional "fractures"?

Answers to Point 7: As also stated in the original text: the open hole section of the wells has been integrated as one dimensional finite elements and homogenisation of the resulting governing equations is done by considering the surface area of the well bore as scaling parameter.

Point 8: It is unclear to me which portions of your work are extensions to MOOSE, and which are already present. This is important and needs to be stated clearly.

Answer to Point 8: MOOSE is a framework, that is, it provides libraries and basic architecture for code development. All the physics described in the manuscript has been implemented by the two authors and we did not make any use of available physical modules as provided alongside the framework.

Point 9: In most reservoirs I have worked with, permeability change is dominated by the behavior of fractures, not by porous mechanics (i.e. Kozeny-Carman), which are minimally important. Perhaps this reservoir is not highly fractured, or perhaps you only interesting in displaying porous behavior for the sake of illustration. In either case, you need to clarify the intent of your assumptions and their shortcomings.

Answer to Point 9: The reviewer might have a point while saying that permeability in reservoir likely is of secondary types, being associated with natural or induced fracturing. We have indeed demonstrate the relevance of multi-frac systems in our first reservoir applications. However, we do not fully agree with the second remark from the reviewer, that is, that the porous reservoir domain is of minimal importance. This is especially not true in sandy reservoirs (see for example Blöcher et al., 2016, computer and geosciences and Jacquey et al., 2016, Tectonophysics) where stimulation mainly target the near well bore area with the aim to increase the volume of fluid exchange to the well, while making use of the natural (porous) permeability of the sandy reservoir compartment to hydraulically connect the injection and production sources.

Point 10: My last and largest comment is in agreement with the previous review. Essentially, you have provided details of capabilities that are then never utilized in the final analysis. I understand this temptation, but it detracts from the story you are trying to tell. The only usage of plasticity in the paper is to demonstrate an oedometer problem. Either add greater complexity to your final problem (I understand there is a fair amount required), or remove plasticity from the discussion. Alternatively, you might choose a validation more along the lines of a Mandel-Cryer type problem (preferably at interesting temperature and pressure and with properties computed internally), which is a fundamental aspect of your final elastic simulation and rounds out the other validations.

Answer to Point 10: Please also refer to our comments to Reviewer#1 as well. Concerning his first remark, we would like to acknowledge the new structure of the examples, following Reviewer#1 comments, as well as the new example dealing with poroelastic and plastic coupling. Though acknowledging the comments from the reviewer, we are a bit puzzled by his second remark. The Mandel-Cryer effect is intended to describe a non-monotonic pore pressure evolution in response to loading or to a change in the stress conditions (where additional, though secondary thermal effects can be implemented). However, to the best of our knowledge it resolves for a poro-(thermal-)elastic medium. By augmenting the original problem with non-constant, p-T dependent properties will of course add to the complexity, but it will also hinder from having any analytics to validate the numerical solution. Therefore, in our opinion, such an example would have the same level of validation as the 3D example we presented as our last study case in the reservoir application part of the results section, which in turn adds complexity on the model geometry (having discrete fractures, one of the peculiar aspect of our approach). In addition, we made reference in the text to a manuscript, which is currently under review, where we investigate the effects of poro-elastic response in a real reservoir due to stimulation activities, that is, a real case 3D Mandel-Cryer problem.